# Outlier Robust Training of Machine Learning Models

## Abstract

Robust training of machine learning models in the presence of outliers has garnered attention across various domains. Use of robust losses is a popular approach and has been known to mitigate the effects of outliers. We bring to light two literatures that have diverged in their ways of designing robust losses: one using robust estimation, which is popular in robot and computer vision, and another using risk-minimization framework, which is popular in deep learning. We *first* show that a simple modification of the Black-Rangarajan duality provides a unifying view. The modified duality brings out a definition of a *robust loss kernel* $\sigma$ that is satisfied by robust losses in both the literatures. *Secondly,* using the modified duality, we propose two classes of algorithms, namely, graduated non-convexity and adaptive training algorithms. These algorithms are augmented with a novel parameter update rule by interpreting the weights in the modified duality as inlier probabilities. *Thirdly*, we investigate convergence of the two algorithms to the outlier-free optima, *i.e.,* the ground-truth. Considering arbitrary outliers (*i.e.,* with no distributional assumption on the outliers), we show that the use of robust loss kernel $\sigma$ increases the region of convergence. We experimentally show the efficacy of our algorithms on regression, classification, and neural scene reconstruction.

## 1 Introduction

Humans are good at detecting outliers and segregating them (Chai et al., 2020). This is not the case when it comes to training machine learning models (Sukhbaatar et al., 2015; Wang et al.; Sabour et al., 2023). Robustly training deep learning models, in presence of outliers, is important. It can off-set the high cost of obtaining accurate annotations. Many works now implement automatic or semi-automatic annotation pipelines which can be leveraged to train models (Armeni et al., 2016; Chang et al., 2017; Tkachenko et al., 2020; Yang et al., 2021; Gadre et al., 2023) A robot that can self-train its internal models by collecting and self-annotating data has been envisioned (Schmidt & Fox, 2020; Deng et al., 2020; Lu et al., 2022; Talak et al., 2023; Shi et al., 2023; Jawaid et al., 2024; Wang et al., 2024), but is still a work in progress.

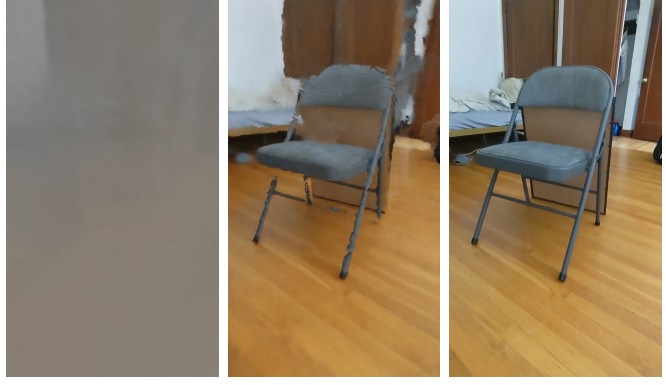

Figure 1: Nerfacto reconstruction results after 80% of the training pixels have been perturbed by outliers: (left) training with the originally proposed Adam; (middle) training with our Adaptive Training Algorithm with Truncated Loss; (right) ground truth.

Robust estimation (Huber, 1981) suggests that in presence of outliers, one should optimize:

$$\underset{\boldsymbol{w} \in \mathsf{W}}{\text{Minimize}} \quad \sum_{i=1}^{n} \rho(f_i(\boldsymbol{w})), \tag{1}$$

instead of

$$\underset{\boldsymbol{w} \in \mathsf{W}}{\text{Minimize}} \quad \sum_{i=1}^{n} f_i(\boldsymbol{w}), \tag{2}$$

where $f_i$ denotes the training loss for measurement $i$. Here, $\rho$ is the robust loss that typically dampens the high loss (*i.e.,* high $f_i(\boldsymbol{w})$) terms. Many robust losses have been proposed in the literature to mitigate the effects of outliers. Recent works in robust estimation in robotics have shown that using a parameterized robust loss $\rho$, with adaptive parameter tuning during training, yields better outlier mitigation (see Sec. 8.2). Many robust losses have also been proposed in training deep learning models for the task of multi-label classification (see Sec. 8.1). However, we observe a divergence in principles that govern the design of robust losses in (a) robotics and computer vision, that uses robust estimation frameworks, and in (b) training deep learning models, that uses risk-minimization frameworks (see Section 2).

Robust estimation as applied in robotics and computer vision first shows that the problem (1) can be written down as a weighted least squares problem

$$\underset{\boldsymbol{w} \in \mathsf{W}, \ u_i \in [0,1]}{\text{Minimize}} \quad \sum_{i=1}^{n} u_i \cdot f_i^2(\boldsymbol{w}) + \Psi_\rho(u_i), \tag{3}$$

where $\Psi_\rho(u)$ is an outlier process that is determined by the Black-Rangarajan duality (Black & Rangarajan, 1996). The equivalence between (1) and (3) is useful for robotics and computer vision applications as all robust estimation problems can be re-written as weighted non-linear least squares, which are typically easy to solve. However, this framework cannot be applied directly to machine learning problems. For example, if $f_i(\boldsymbol{w})$ is the cross-entropy loss, minimizing the squared cross-entropy loss does not make an equal sense.

Ghosh et al. (2015; 2017) develops the notion of noise-tolerant loss using risk minimization framework. Let model weight $\boldsymbol{w}_\lambda$ minimizes risk when there are $\lambda$ fraction of outliers. Ghosh et al. (2015; 2017) say that the loss used in defining the risk is noise-tolerant provided $\boldsymbol{w}_\lambda = \boldsymbol{w}_0$ (*i.e.,* equal to the optimal weights $\boldsymbol{w}_0$ when there are no outliers). Several noise-tolerant losses have been proposed since then that have shown improved performance at mitigating the presence of outliers in the training data. These losses include generalized cross entropy, symmetric cross entropy, reverse cross entropy, Taylor cross entropy, among others (see Section 8.1). While the setup has some advantages it suffers from some limitations; for instance, one has to assume an outlier distribution to derive the noise-tolerant loss. As an instance of triviality that results from this, one can show that mean square error (MSE) loss is noise-tolerant under zero mean outliers. However, it is well known that MSE is not robust for finite sample problems (*i.e.,* $n$ in (2) is finite), and even one outlier can significantly degrade the estimate significantly (Huber, 1981).

Algorithms have been proposed to train deep learning models in presence of outliers in the training data (see Section 8.1 and 8.3). These have been either the heuristic approaches applied on the specific task of multi-label classification (Elesedy & Hutter, 2023; Li et al., 2020), or algorithms for solving a general stochastic optimization problem, albeit with outliers (Menon et al., 2020; Merad & Gaïffas, 2024; Chhabra et al., 2024; Hu et al., 2024; Shen & Sanghavi, 2019; Shah et al., 2020; Prasad et al., 2020). The heuristic methods do not have a strong theoretical grounding. The methods based on stochastic optimization analyze their convergence properties assuming an outlier distribution. To the best of our knowledge, there are no works that analyze the region of convergence for stochastic gradient-based algorithm under arbitrary outliers.

## 1.1 Contribution

This paper makes the following key contributions:

1. We expose two divergent approaches to designing robust loss functions, for training machine learning models, in the presence of outliers. The first, based on robust estimation framework, and the second,

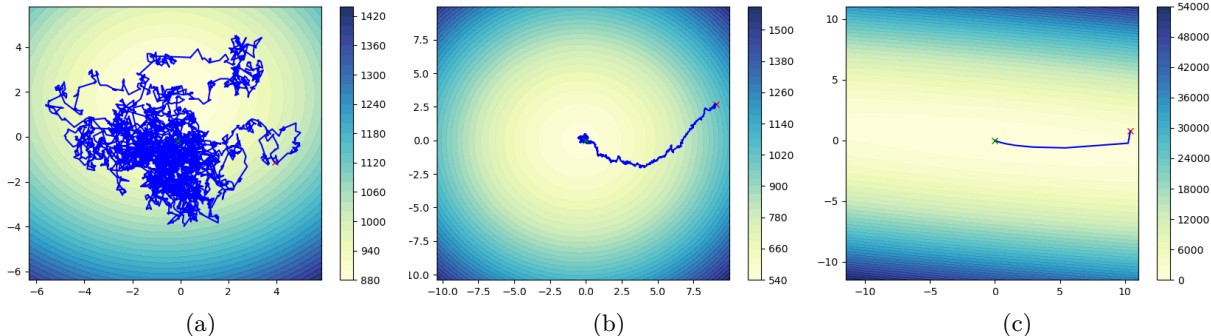

Figure 2: Shows trajectory of (a) SGD (batch size = 1), (b) Adaptive Algorithm with Linear Truncated Loss (batch size = 1), and (c) Gradient Descent, for a linear regression problem with zero-mean outliers. The presence of outliers in the training data makes the SGD deviate. The outlier measurements arbitrarily perturb it's gradient estimate. Adaptive and GNC algorithms stabilize the descent and the variance in the gradient estimate is lower (Lemma 14). We observe its behavior to be close to the full gradient descent, where the gradient estimate is exact, given zero-mean outliers.

based on risk minimization framework. We highlight that the Black-Rangarajan duality-based robust estimation framework is not directly applicable to the machine learning contexts, as it reformulates the M-estimation problem as a weighted sum of least squares.

2. We prove a simple modification of the Black-Rangarajan duality that preserves the problem structure and makes it applicable to machine learning problems. The modification ensures that the square term $f_i^2(\boldsymbol{w})$ in (3) becomes linear, *i.e.,* $f_i(\boldsymbol{w})$. Most importantly, the modified Black-Rangarajan duality gives rise to a definition of a robust loss kernel $\sigma$. We show that this robust loss kernel unifies the robust losses used in the two literatures of robust estimation and training deep classifiers. This enables one to now use the robust loss kernels, developed in the deep learning literature, in the robust estimation problems, and vice versa.

3. We introduce two classes of training algorithms based on the modified Black-Rangarajan duality: graduated non-convexity and adaptive training algorithms. We propose a percentile rule for parameter update, thereby eliminating the need for parameter tuning. We show connections between these algorithm classes and prior work. We also show that the parameter update rule can be interpreted as training on conformal prediction sets, generated during training.

4. Our main contribution is the convergence analysis for the two training algorithms. Under mild assumptions, we demonstrate that (i) the robust loss kernel $\sigma$ in both algorithms expands the region of convergence compared to vanilla stochastic gradient descent on the original objective, and (ii) the use of $\sigma$ reduces variance in iterates, enabling stable descent and improved convergence (an instance shown in Figure 2). We validate these findings experimentally on linear regression and multi-label classification tasks. We demonstrate the algorithms' efficacy in mitigating pixel-level outliers in neural scene rendering (Nerfacto; Tancik et al. (2023)), successfully recovering images with up to 80% outliers, shown in Figure 1.

## 1.2 Organization

The paper is organized as follows: Section 2 provides the background and elucidates the divergent perspectives on robust loss design. Section 3 delineates the problem setup. Section 4 introduces the modified Black-Rangarajan duality and the unified robust loss kernel $\sigma$. Section 5 presents the two class of algorithms, and Section 6 discusses their convergence. Section 7.2 reports the experimental findings. Section 8 discusses related work, and Section 9 concludes the study. All mathematical proofs are provided in the Appendices

## 2 Background: Diverging Principles of Robust Loss Design

We first review the principles that govern robust loss design in: (i) robust estimation in robot and computer vision (Section 2.1), and (ii) training deep learning in presence of outliers (Section 2.2). We show the contrast between these two views and the need to reconcile them.

### 2.1 Robust Estimation in Robot and Computer Vision

Many problems in robot and computer vision are formulated as least squares problems:

$$\underset{\boldsymbol{w}\in\mathsf{W}}{\text{Minimize}} \quad \sum_{i=1}^{n} r_i^2(\boldsymbol{w}), \tag{4}$$

where $r_i(\boldsymbol{w})$ denote the residual error on measurement $i$ and is typically a non-linear function of model weights $\boldsymbol{w}$. This makes solving (4) hard. This difficulty is exacerbated when the measurements contain outliers. In presence of outliers, the global optima of (4) can deviate considerably from the ground-truth $\boldsymbol{w}^*$. Robust estimation is used to address this issue by re-formulating (4) as an M-estimator:

$$\underset{\boldsymbol{w}\in\mathsf{W}}{\text{Minimize}} \quad \sum_{i=1}^{n} \rho(r_i(\boldsymbol{w})), \tag{5}$$

where $\rho$ is a robust loss function.[1] Many robust losses have been proposed in the literature including truncated least squares loss, Geman McClure loss, Welsch-Leclerc loss, Cauchy-Lorentzian loss, Charbonnier loss. Barron (2019) proposes a parameterized robust loss that equal many robust losses upon appropriate choice of the parameters.

**Truncated Least Square loss.** $\rho(r) = \min\{r^2, c^2\}$.

**Geman McClure loss.** $\rho(r) = \frac{c^2 r^2}{c^2 + r^2}$.

**Welsch-Leclerc loss.** $\rho(r) = 1 - \exp(-\frac{1}{2} r^2/c^2)$.

**Cauchy-Lorentzian loss.** $\rho(r) = \log(1 + \frac{1}{2} r^2/c^2)$.

**Charbonnier loss.** $\rho(r) = \sqrt{r^2/c^2 + 1} - 1$.

**Barron's loss.** $\rho(r) = \frac{|\alpha-2|}{\alpha} \left( \left( \frac{(x/c)^2}{|\alpha-2|} + 1 \right)^{\alpha/2} - 1 \right)$

The robust estimation problem (5) is solved by re-writing it as a weighted least squares problem:

$$\underset{\boldsymbol{w}\in\mathsf{W}, u_i\in[0,1]}{\text{Minimize}} \quad \sum_{i=1}^{n} u_i r_i^2(\boldsymbol{w}) + \Psi_\rho(u_i), \tag{6}$$

where $\Psi_\rho$ is an outlier process (*i.e.,* function that depends on the robust loss function $\rho$). Black-Rangarajan duality (Black & Rangarajan, 1996) shows the equivalence between the M-estimator (5) and the weighted non-linear least squares problem (6).

**Theorem 1** (Black & Rangarajan (1996))**.** *The robust estimation problem* (5) *is equivalent to the weighted non-linear least squares problem* (6) *with* $\Psi_\rho(u) = -u(\phi')^{-1}(u) + \phi((\phi')^{-1}(u))$ *and* $\phi(r) = \rho(\sqrt{r})$, *provided* $\phi(r)$ *satisfies: (i)* $\phi'(r) \to 1$ *as* $r \downarrow 0$, *(ii)* $\phi'(r) \to 0$ *as* $r \uparrow 1$, *and (iii)* $\phi''(r) < 0$.

---

[1]We use the notation $\rho$ here and keep $\sigma$ for the unified robust loss kernel defined in Section 4.

Black-Rangarajan duality motivates solving (5) by iteratively minimizing the weighted non-linear least squares problem. The coefficient weights $u_i = \rho(r_i(\boldsymbol{w}))/2r_i(\boldsymbol{w})$ are chosen using $\boldsymbol{w}$ from the previous iteration (Black & Rangarajan, 1996). This has been leveraged to develop robust algorithms for estimation problems in computer vision and robotics (*e.g.,* Yang et al. (2020a); Chebrolu et al. (2020); Peng et al. (2023)).

## 2.2 Training Deep Learning Models in Presence of Outliers

Risk-minimization framework suggests that a deep learning model is trained to obtain the model weights:

$$\boldsymbol{w}^* = \arg\min_{\boldsymbol{w} \in W} \ \mathbb{E}_{(\boldsymbol{x},\boldsymbol{y}) \sim \mathcal{D}}[l(\boldsymbol{g}(\boldsymbol{w},\boldsymbol{x}), \boldsymbol{y})], \tag{7}$$

where $l : Y \times Y \to \mathbb{R}_+$ be a loss function, $Y$ denote the set of all outputs, $\mathcal{D}$ denotes the distribution of pairs $(\boldsymbol{x}, \boldsymbol{y})$ when there are no outliers, and $\boldsymbol{g}(\boldsymbol{w}, \boldsymbol{x})$ is the model that predicts output $\boldsymbol{y}$, given input $\boldsymbol{x}$ and model weights $\boldsymbol{w}$. The goal of robust loss $l$ design should be such that $\boldsymbol{w}^*$ does not change much when we introduce outliers in the distribution $\mathcal{D}$. Let

$$\boldsymbol{w}_\lambda^* = \arg\min_{\boldsymbol{w} \in W} \ \mathbb{E}_{(\boldsymbol{x},\boldsymbol{y}) \sim \mathcal{D}_\lambda}[l(\boldsymbol{g}(\boldsymbol{w},\boldsymbol{x}), \boldsymbol{y})], \tag{8}$$

denote the optimal model weights when the dataset contains $\lambda$ fraction of outliers; here, training data now comes from outlier-infested distribution $\mathcal{D}_\lambda$, where $\lambda$ fraction of data are outliers. A loss function is said to be *noise-tolerant* at noise rate $\lambda$ if $\boldsymbol{w}^* = \boldsymbol{w}_\lambda^*$.

Ghosh et al. (2015; 2017) show that the classical cross entropy (CE) loss is not noise-tolerant for the task of classification. It further shows that a simple mean absolute error (MAE) loss is noise-tolerant to any $\lambda < 1 - 1/K$ fraction of outliers, where $K$ denote the total number of classes. Several noise-tolerant losses have been proposed since including generalized cross entropy (GEC) loss, symmetric cross entropy loss (SCE), finite Taylor series expansion of log likelihood loss, and asymmetric losses (see Section 8.1). Let $\boldsymbol{p} = \boldsymbol{g}(\boldsymbol{w}, \boldsymbol{x})$ and $\boldsymbol{p}[y]$ denote the predicted probability of class label $y$:

**Mean absolute error (MAE).** $l(\boldsymbol{p}, y) = 1 - \boldsymbol{p}[y]$.

**Generalized cross entropy (GCE).** $l(\boldsymbol{p}, y) = \frac{1}{q}(1 - \boldsymbol{p}[y]^q)$.

**Symmetric cross entropy (SCE).** $l(\boldsymbol{p}, y) = -\log(\boldsymbol{p}[y]) - A\sum_{k \neq y} \boldsymbol{p}[k]$.

**Reverse cross entropy (RCE).** $l(\boldsymbol{p}, y) = -A\sum_{k \neq y} \boldsymbol{p}[k]$.

**Taylor cross entropy (t-CE).** $l(\boldsymbol{p}, y) = \sum_{m=1}^t \frac{1}{m}(1 - \boldsymbol{p}[y])^m$.

**Asymmetric generalized cross entropy (AGCE).** $l(\boldsymbol{p}, y) = \frac{1}{q}\left((a+1)^q - (a+\boldsymbol{p}[y])^q\right)$.

**Asymmetric unhinged loss (AUL).** $l(\boldsymbol{p}, y) = \frac{1}{p}\left((a-\boldsymbol{p}[y])^p - (a-1)^p\right)$.

**Asymmetric exponential loss (AEL).** $l(\boldsymbol{p}, y) = \exp\left(-\boldsymbol{p}[y]/a\right)$.

All these losses are up to a constant away from the original losses in the literature (Zhang & Sabuncu, 2018; Amid et al., 2019; Wang et al., 2019; Feng et al., 2020; Zhou et al., 2023). We take this liberty because a constant factor does not affect the optima. Note that these losses can be written down as

$$l(\boldsymbol{p}, y) = \rho(-\log \boldsymbol{p}[y]), \tag{9}$$

where $-\log \boldsymbol{p}[y]$ is the standard cross entropy loss. This implies that we can construct a robust loss kernel $\rho$ for each of these losses with respect to the standard cross entropy loss. However, several problems arise

in articulating the above losses this way. Firstly, a direct application of Black-Rangarajan duality yields an equivalence between the robust estimation problem as the squared cross entropy loss:

$$\underset{\boldsymbol{w}\in\mathsf{W}, u_i\in[0,1]}{\text{Minimize}} \quad \sum_{i=1}^{n} u_i \left(\log(\boldsymbol{p}[y])\right)^2 + \Psi_\rho(u_i), \tag{10}$$

which doesn't make much sense. Ideally, we would like a duality result where the robust estimation problem is shown to be equivalent to a weighted cross entropy minimization problem in (10). In general, the dual of the robust estimation should be a weighted version of the original problem. The weights should indicate the confidence in the sample being an outlier. We would then be able to apply this result to non-linear least squares, as well as cross-entropy minimization.

The second problem that arises in using the dual (10) is that many of the robust loss kernels $\rho$ (see (9)) used in this conversion do not satisfy the requirement on $\rho$ in Theorem 1.

**Remark 2** (Risk Minimization Framework and Robust Losses). *The risk minimization framework used to define robust loss (*i.e.*, noise-tolerant loss) is limiting. The framework ignores the effects of finite sample size and forces the designer to make unrealistic assumptions about outlier distribution. For example, it can be easily shown that the mean squared error (MSE) is noise-tolerant for regression problems, if the outliers are zero mean. However, fragility of MSE to outliers in finite sample problems is well documented (Huber, 1981). Figure 2 shows that even when training using the MSE loss (which is noise-tolerant) using SGD results in convergence issues due to high variance in gradient estimates.*

**Remark 3** (Convergence and Robust Loss Design). *While many works have proposed robust losses for training deep learning models, there has been a little effort at understanding the effect of robust losses on convergence in presence of outliers. Analytical results relating the structure of the robust loss to region of convergence and outlier mitigation are unknown. A lack of any structure on the robust losses $\rho$ mean that researchers have to heuristically guess and experiment on a wide variety of datasets before being confident.*

## 3 Problem Statement

We are given a dataset of $n$ samples which contain outliers. Each sample is an input-output pair $(\boldsymbol{x}_i, \boldsymbol{y}_i)$. The goal is to train a model while mitigating the presence of outliers in the dataset. The model is parameterized by model weights $\boldsymbol{w} \in \mathbb{R}^d$. In the outlier-free case, each measurement $i$ is associated with a loss $f_i(\boldsymbol{w}) = l(\boldsymbol{h}(\boldsymbol{w}, \boldsymbol{x}_i), \boldsymbol{y}_i) \geq 0$, and the model is trained to solve the following optimization problem:

$$\underset{\boldsymbol{w}\in\mathsf{W}}{\text{Minimize}} \quad f(\boldsymbol{w}) = \frac{1}{n}\sum_{i=1}^{n} f_i(\boldsymbol{w}). \tag{11}$$

When the measurements are infested with outliers, we would ideally like to minimize the following, outlier-free objective, instead of (11):

$$\underset{\boldsymbol{w}\in\mathsf{W}}{\text{Minimize}} \quad f_I(\boldsymbol{w}) = \frac{1}{n}\sum_{i=1}^{n} f_{i,I}(\boldsymbol{w}), \tag{12}$$

where $f_{i,I}(\boldsymbol{w})$ denote the outlier-free component of the loss, *i.e.,* as if the outlier was completely removed from the measurement. However, it is not possible to know $f_{i,I}(\boldsymbol{w})$ in practice and we are constrained to work with $f(\boldsymbol{w})$ and $f_i(\boldsymbol{w})$, while attempting to minimize (12). Let $f_I^*$ denote the optimal value of (12). The goal is to find $\hat{\boldsymbol{w}}$ such that $f_I(\hat{\boldsymbol{w}})$ is as close to $f_I^*$ as possible, and we need to do this using only $f_i(\boldsymbol{w})$ and $f(\boldsymbol{w})$ in (11). We do not know $n_O$, the number of outliers, and assume that they are arbitrary, and do not follow a specific distribution. We use $n_I = n - n_O$ and $\lambda = n_O/n$ to denote the number of inliers and the fraction of outlier measurements, respectively.

In the next section, we bring out a unified definition of a robust loss kernel $\sigma$ (Definition 6) based on a simple modification of the Black-Rangarajan duality (Corollary 4). In Section 5, we make use of the modified Black-Rangarajan duality to propose two training algorithms, for training deep learning models, in presence of arbitrary outliers. We prove convergence properties of these algorithms in Section 6.

# 4 Unified Robust Loss Kernel

We now present the unified framework. We first prove a modified version of the Black-Rangarajan duality. This version helps keep the problem structure intact. That is, the dual of a robust cross entropy minimization problem is a weighted cross entropy minimization problem. Similarly, the dual of a robust non-linear least squares estimation problem is a weighted non-linear least squares problem. In both cases, weights indicate the confidence that the measurement is an inlier (*i.e.,* higher the weight, greater the confidence that the measurement is an inlier). The modified dual also gives rise to a definition of the robust loss kernel $\sigma$ that is simple, intuitive, and verifiable. We will see that all the robust losses we have seen in Sections 2.1-2.2 can be modified to meet this definition.

## 4.1 Modified Black-Rangarajan Duality

We state and prove a modified version of the Black-Rangarajan Duality.

**Corollary 4** (Modified Black-Rangarajan Duality). *The robust estimation problem,*

$$\underset{\boldsymbol{w}, u_i \in [0,1]}{Minimize} \quad \frac{1}{n} \sum_{i=1}^{n} \sigma(f_i(\boldsymbol{w})), \tag{13}$$

*with robust loss kernel $\sigma(\cdot)$ is equivalent to*

$$\underset{\boldsymbol{w}, u_i \in [0,1]}{Minimize} \quad \frac{1}{n} \sum_{i=1}^{n} \left[ u_i \cdot f_i(\boldsymbol{w}) + \Phi_\sigma(u_i) \right], \tag{14}$$

*where $\Phi_\sigma(u) = -u(\sigma')^{-1}(u) + \sigma((\sigma')^{-1}(u))$, provided $\sigma(r)$ satisfies: (i) $\sigma'(r) \to 1$ as $r \downarrow 0$, (ii) $\sigma'(r) \to 0$ as $r \uparrow 1$, and (iii) $\sigma''(r) < 0$.*

**Proof.** The proof is obtained by substituting $\sigma(r) = \rho(\sqrt{r})$ and $r_i(\boldsymbol{w})^2 = f_i(\boldsymbol{w})$ in the Black-Rangarajan duality (Theorem 1). We provide a proof from first principles in Appendix A. $\qquad\square$

**Remark 5** (Dual Problem Structure and its Application). *The modified Black-Rangarajan duality keeps the problem structure intact,* i.e., *the dual problem minimizes a sum of weighted losses $f_i(\boldsymbol{w})$. This is in contrast with the original Black-Rangarajan dual where the dual problem would have been to minimize the sum of weighted squares $f_i^2(\boldsymbol{w})$. This allows us to apply the modified Black-Rangarajan duality to train deep learning models in presence of outliers (see Section 5).*

## 4.2 Unified Robust Loss Kernel

The modified Black-Rangarajan duality imposes constraints on $\sigma$. We investigate these constraints and see that it forms a simple, intuitive, and verifiable definition of a *robust loss kernel* that can be applied generally across all deep learning problems. The modified duality (Corollar 4) requires:

C1: $\sigma'(r) \to 1$ and $r \downarrow 0$.

C2: $\sigma'(r) \to 0$ and $r \uparrow +\infty$.

This indicates that $\sigma$ should be such that for small $r$ it behaves like a linear function, *i.e.,* $\rho(r) \approx r$ for $r$ close to 0. For large $r$, on the other hand, $\sigma$ behaves like a constant function. As a robust loss kernel, for smaller loss terms, it pretends to not exist, while for larger loss terms, it damps their effect on the total loss. The third condition:

C3: $\sigma''(r) < 0$,

Table 1: Robust loss kernels that correspond to popular robust losses in robust estimation in robot and computer vision (Section 2.1) and in training deep learning models (Section 2.2).

| Robust Loss Kernel | $\sigma(r)$ |
| --- | --- |
| Linear Truncated Kernel | $c \cdot \min\{r/c, 1\}$ |
| Geman McClure Kernel | $c \cdot \frac{r/c}{1+r/c}$ |
| Welsch-Leclerc Kernel | $c \cdot (1 - \exp(-r/c))$ |
| Cauchy-Lorentzian Kernel | $c \cdot \log(1 + r/c)$ |
| Charbonnier Kernel | $2c \cdot \sqrt{r/c + 1} - 1$ |
| Barron's Kernel | $c \cdot \frac{|\alpha-2|}{\alpha} \left( \left( \frac{r/c}{|\alpha-2|} + 1 \right)^{\alpha/2} - 1 \right)$ |
| Mean error kernel | $1 - \exp(-r)$ |
| Generalized cross entropy kernel | $\frac{1}{q}(1 - \exp(-qr))$ |
| Symmetric cross entropy kernel | $\frac{1}{1+A}(r + A\exp(-r) - A)$ |
| Taylor cross entropy kernel | $\sum_{m=1}^{t} \frac{1}{m}(1 - \exp(-r))^m$ |
| Asymmetric generalized cross entropy kernel | $\frac{1}{q \cdot a^{q-1}}((a+1)^q - (a + \exp(-r))^q)$ |
| Asymmetric unhinged kernel | $\frac{1}{p \cdot a^{p-1}}((a - \exp(-r))^p - (a-1)^p)$ |
| Asymmetric exponential loss | $a \cdot \exp\left(\frac{1}{a}(1 - \exp(-r))\right)$ |

implies that $\sigma'(r)$ is a monotonically decreasing function and $\sigma'(r) \in [0, 1]$ for all $r$. A consequence of this is that $\sigma$ is a monotonically increasing function, and therefore, preserves ordering of the losses (*i.e.,* $f_i(\boldsymbol{w}) \le f_j(\boldsymbol{w})$ implies $\sigma(f_i(\boldsymbol{w})) \le \sigma(f_j(\boldsymbol{w}))$). All this makes for a simple, intuitive, and verifiable definition of a robust loss kernel $\sigma$:

**Definition 6** (Robust Loss Kernel $\sigma$). *A function $\sigma : \mathbb{R} \to \mathbb{R}$ is a robust loss function if (i) $\sigma'(r) \to 1$ as $r \downarrow 0$, (ii) $\sigma'(r) \to 0$ as $r \uparrow +\infty$, and (iii) $\sigma''(r) \le 0$.*

We relax the strict concavity of $\sigma$ in C3 to the condition $\sigma''(r) \le 0$. The strict concavity is required for the modified Black-Rangarajan duality to hold (in particular, to ensure invertibility of $\sigma'$). A truncated kernel $\sigma(r) = c \min\{r/c, 1\}$ does not have an invertible $\sigma'$ (and does not satisfy C3), but can still be a valid robust loss kernel according to Definition 6.

Table 1 presents various robust loss kernels. The first six robust kernels are derived from the robust losses in the robust estimation in robot and computer vision literature (see Section 2.1). The next eight kernels are derived from the robust losses used in training deep learning-based classifier models (see Section 2.2). It can be verified that each robust loss kernel corresponds to a robust loss presented in Sections 2.1-2.2 (see Appendix B). All the kernels presented in Table 1 satisfy Definition 6, and can be applied with the modified Black-Rangarajan duality on various machine learning problems.

# 5 Robust Training Algorithms

We present two algorithms for training machine learning models in presence of outliers. Both these algorithms are a consequence of the modified Black-Rangarajan duality. In Section 6, we will analyze convergence properties of these algorithms and demonstrate their usefulness in Section 7.

## 5.1 Graduated Non-Convexity Algorithms

The modified Black-Rangarajan duality (Corollary 4) motivates our *graduated non-convexity algorithm*. The key idea is to solve the dual (*i.e.,* (14)). We do this by an alternating minimization strategy, in which we we first optimize/update the model weights $\boldsymbol{w}$, given $\boldsymbol{u}$, and then optimize/update $\boldsymbol{u}$, given $\boldsymbol{w}$. We use a robust loss kernel $\sigma_c$ that is parameterized by a constant $c$. This parameter gives us additional flexibility as we can modulate $\sigma_c$ as the training progresses. We update $c$ as the training iterations progress.

**Algorithm 1** (Graduated Non-Convexity Training Algorithm). *Starting with initial model weights $\boldsymbol{w}_0$, coefficient weights $u_{i,0}$, and the robust loss kernel's parameter $c_0$, we update the coefficient weights as*

$$u_{i,t+1} = \underset{u\in[0,1]}{ArgMinimize}\ \ u \cdot f_i(\boldsymbol{w}_t) + \Phi_{\sigma_{c_t}}(u), \tag{15}$$

*and the model weights as*

$$\boldsymbol{w}_{t+1} = \underset{\boldsymbol{w}}{ArgMinimize}\ \frac{1}{n}\sum_{i=1}^{n} u_{i,t} \cdot f_i(\boldsymbol{w}). \tag{16}$$

*The weight update step can be performed in many ways. It can involve running any existing gradient-based algorithm $\mathcal{A}$ (e.g., SGD, ADAM) either to convergence or running it for a few iterations. The parameter $c_t$ is updated every few iterations according to an update rule in Section 5.3.*

We next show that (15) has a simple analytical solution. The proof is given in Appendix C.

**Lemma 7.** *The $u^*$ that solves $\underset{u\in[0,1]}{\arg\min} uf_i(\boldsymbol{w}) + \Phi_{\sigma_c}(u)$ is given by $u^* = \sigma_c'(f_i(\boldsymbol{w}))$.*

This simplifies the implementation of Algorithm 1, as the weight update step reduces to

$$\boldsymbol{w}_{t+1} = \underset{\boldsymbol{w}}{\text{ArgMinimize}}\ \ \frac{1}{n}\sum_{i=1}^{n}\sigma_{c_t}'(f_i(\boldsymbol{w}_t)) \cdot f_i(\boldsymbol{w}). \tag{17}$$

Note that the coefficient weights $u_{i,t} = \sigma_{c_t}'(f_i(\boldsymbol{w}_t))$ remain fixed and is determined by $\boldsymbol{w}_t$.

## 5.2 Adaptive Training Algorithms

The graduated non-convexity algorithm iteratively computes coefficient weights from measurements and solves the optimization problem (17) to update the model weights. If the weight update (17) is performed using a one step of a gradient-based algorithm (*e.g.,* SGD, ADAM), the gradient looks like

$$\boldsymbol{g}_t = \frac{1}{|\mathcal{B}|}\sum_{i\in\mathcal{B}}\ \sigma_{c_t}'(f_i(\boldsymbol{w}_t)) \cdot \nabla f_i(\boldsymbol{w}_t), \tag{18}$$

where $\mathcal{B}$ denotes the training batch. In a hypothetical setting, if the batch size was the full set (*i.e., $\mathcal{B} = [n]$*), $\boldsymbol{g}_t$ would be the gradient of the objective in the robust estimation problem:

$$\underset{\boldsymbol{w},u_i\in[0,1]}{\text{Minimize}}\ \ \frac{1}{n}\sum_{i=1}^{n}\sigma_{c_t}(f_i(\boldsymbol{w})), \tag{19}$$

The only difference is that the parameter $c_t$ gets updated rather than staying constant as in (13). This motivates our adaptive training algorithm.

**Algorithm 2** (Adaptive Training Algorithms). *Adaptive training algorithms are gradient-based algorithms that use the gradient*

$$\boldsymbol{g}_t = \frac{1}{|\mathcal{B}|} \sum_{i \in \mathcal{B}} \sigma'_{c_t}(f_i(\boldsymbol{w}_t)) \cdot \nabla f_i(\boldsymbol{w}_t), \tag{20}$$

*if the gradient is computed using a batch $\mathcal{B}$ of samples. At every training iteration $t$, model weights are updated using any of the existing gradient-based algorithms $\mathcal{A}$ (e.g., SGD, ADAM). The robust loss parameter $c_t$ is updated every few iterations according to an update rule in Section 5.3.*

The algorithm is called *adaptive* to emphasize that the robust kernel parameter is updated as the training progresses. In the next section, we give the parameter update rule.

### 5.3 Parameter Update

**Algorithm 3** (Parameter Update Rule). *Given model weights $\boldsymbol{w}_t$ and the parameterized robust loss kernel $\sigma_c$, the parameter update $c_{t+1}$ is computed by solving*

$$c_{t+1} = \underset{c \in [0,1]}{Find} \left\{ \frac{1}{|\mathcal{D}|} \sum_{i \in \mathcal{D}} \sigma'_c(f_i(\boldsymbol{w}_t)) = \zeta \right\}, \tag{21}$$

*where $\mathcal{D}$ denotes the set of all accumulated measurements across previous iterations (i.e., with $c = c_t$) and $\zeta$ is a positive constant.*

The rational for this rule is as follows. In the dual problem (17), the coefficient weights $\sigma'_c(f_i(\boldsymbol{w}_t))$ can be interpreted as the likelihood that the measurement $i$ is an inlier, *i.e.*, $\mathbb{P}(i \in n_I \mid \boldsymbol{w}_t, \boldsymbol{c})$. However, we note that these probabilities should satisfy a constraint. Note that there are a fixed number of outliers and inliers, respectively; *i.e.*, $\sum_{i=1}^{n} \mathbb{I}\{i \in n_I\} = n_I$. Taking conditional expectation on both sides (w.r.t. $\boldsymbol{w}_t, \boldsymbol{c}$) we obtain $\sum_{i=1}^{n} \mathbb{P}(i \in n_I \mid \boldsymbol{w}_t, \boldsymbol{c}) = n_I$, which implies

$$\frac{1}{n} \sum_{i=1}^{n} \sigma'_c(f_i(\boldsymbol{w}_t)) = \frac{n_I}{n}. \tag{22}$$

Thus, the average of all the $\sigma'_c(f_i(\boldsymbol{w}_t))$ must be a constant. In fact, we know that it should equal the fraction of outliers in the training data. We impose this constraint to obtain our parameter update. We can tune $\zeta$ as a hyper-parameter. We implement (21) using a simple binary search algorithm.

**Remark 8** (Parameter Update and GNC Algorithms in Robust Estimation). *The GNC algorithms in (Yang et al., 2020a; Chebrolu et al., 2020; Peng et al., 2023) construct auxiliary losses, parameterized by $\mu$, and update $\mu$ so that the parameterized loss converges to $\sigma_c$ as the training progresses. The parameter $c$ is left untouched and is treated as a hyper-parameter. In our case, we do not construct any auxiliary loss, and directly update parameter $c$ in training. The parameter update rule, however, results in another hyper-parameter, i.e., $\zeta$. For a user, $\zeta$ is easier to set as it is related to the fraction of inliers in the training dataset.*

**Remark 9** (Parameter Update and Iteratively Trimmed Loss Minimization). *The parameter update (21) updates the robust loss kernel $\sigma_c$ enabling it to better segregate between inliers and outliers. With this parameter update, the graduated non-convexity training algorithm can be viewed as a generalization of the iteratively trimmed loss minimization by Shen & Sanghavi (2019). In Shen & Sanghavi (2019), they train on the best $\alpha n$ measurements (here, best implies measurements with the lowest loss $f_i(\boldsymbol{w}_t)$). Note that when $\sigma_c(r) = c \cdot \max\{r/c, 1\}$ the update rule (21) becomes*

$$c_{t+1} = Find_{c \in [0,1]} \left\{ \frac{1}{|\mathcal{D}|} \sum_{i \in \mathcal{D}} \mathbb{I}\{f_i(\boldsymbol{w}_t) \le c\} = \zeta \right\}, \tag{23}$$

i.e., *it selects $c_{t+1}$ such that $\zeta n$ best samples are used in training. A different, differentially continuous robust kernel $\sigma_c$, generalizes this rule. This implies that the graduated non-convexity (Algorithm 1) and adaptive training (Algorithm 2) algorithms are a generalization of the iterative trimmed loss minimization proposed by Shen & Sanghavi (2019).*

**Remark 10** (Parameter Update and Conformal Set Prediction)**.** *Note that* (23)*, in fact, generates a conformal prediction set given a quantile $\zeta$ (Shafer & Vovk, 2008). The set $C_{t+1} = \{i \in [n] \mid f_i(\boldsymbol{w}_t) \leq c_{t+1}\}$ is the predicted set of good samples that fall within the $\zeta$ quantile. Using $\sigma_c(r) = c \cdot \min\{r/c, 1\}$, therefore, results in an algorithm where one computes a conformal prediction set of samples, and trains on them. The process iterates till convergence.*

In the next section, we prove convergence results for the graduated non-convexity and adaptive training algorithms that use the parameter update rule.

# 6 Theoretical Analysis

We now show that the graduated non-convexity and adaptive training algorithms (i) reduce the variance in the gradient (Section 6.2) and (ii) increase the region of convergence (*i.e.*, convergence to $f_I^*$), in presence of outliers (Section 6.3). Prior works mostly show convergence to the M-estimator objective. Out goal instead is to see when the model converges to the outlier-free optima, *i.e.*, $f_I^*$.

## 6.1 Assumption on Outliers

We first make a few assumptions about how the outliers impact the outlier-free objective $f_I(\boldsymbol{w})$. It turns out that we do not require an explicit relation between the loss component $f_i(\boldsymbol{w})$ and it's outlier-free version $f_{i,I}(\boldsymbol{w})$. As both graduated non-convexity and adaptive training algorithms are gradient-based, we only require assumption about how the outlier $\boldsymbol{o}_i$ impact the gradient $\nabla f_i(\boldsymbol{w})$. We assume that the outliers perturb the true gradient $\nabla f_{i,I}(\boldsymbol{w})$ in an additive manner.

**Assumption 11** (Outlier Gradient)**.** *For outlier measurements $i \in n_O$ we have*

$$\nabla f_i(\boldsymbol{w}) = \nabla f_{i,I}(\boldsymbol{w}) + \boldsymbol{h}_i(\boldsymbol{o}_i, \boldsymbol{w}), \tag{24}$$

*where $\boldsymbol{h}_i(\boldsymbol{o}_i, \boldsymbol{w}) \in \mathbb{R}^d$ and is unknown.*

We verify that this assumption holds for two broad class of problems, namely non-linear regression and multi-label classification in Appendix D.

**Remark 12** (Huber Contamination Model)**.** *We remark here that Assumption 11 is different from the Humber contamination model considered in recent works (Merad & Gaïffas, 2024; Prasad et al., 2020). In it, the outlier factor $\boldsymbol{h}_i(\boldsymbol{o}_i, \boldsymbol{w})$ does not depend on $i$ and is assumed to follow a distribution. In our study, we consider $\boldsymbol{h}_i(\boldsymbol{o}_i, \boldsymbol{w})$ not only to depend on $i$ and is arbitrary.*

We next make a final assumption to make things analytically easier. However, this assumption could be relaxed to obtain tighter bounds.

**Assumption 13** (Low Signal-to-Outlier Ratio)**.** *Outlier noise is large and is larger than it's signal,* i.e., $\|\boldsymbol{h}_i(\boldsymbol{o}_i, \boldsymbol{w})\| \geq 1$, $\|\boldsymbol{h}_i(\boldsymbol{o}_i, \boldsymbol{w})\| \geq \|\nabla f_{i,I}(\boldsymbol{w})\|$ *for all $i \in n_O$.*

## 6.2 Variance in Updates

Outliers in the dataset can affect the computed gradients $\boldsymbol{g}_t$ and render the algorithm to be unstable and not converge to the optima (see Figure 2). Loss function $f_i(\boldsymbol{w})$ plays a role in determining how the outliers effect the gradients (see Examples 24 and 25). We next show how the adaptive algorithm (Algorithm 2) is able to control the variance of the descent direction better. We consider batch size of one for ease of presentation.

**Lemma 14.** *Consider batch size of one in training algorithms and the outliers to be zero mean,* i.e., $\frac{1}{n_O}\sum_{i \in n_O} \boldsymbol{h}_i(\boldsymbol{o}_i, \boldsymbol{w}) = 0$. *The variance in the descent direction,* i.e., $\mathbb{E}_i[\|\boldsymbol{g}_t - \nabla f_I(\boldsymbol{w})\|^2]$, *for stochastic gradient descent and the adaptive method is bounded by*

$$3\eta^2\lambda\frac{1}{n_O}\sum_{i=1}^{n_O}\|\boldsymbol{h}_i(\boldsymbol{o}_i, \boldsymbol{w}_t)\|^2, \tag{25}$$

*and*

$$3\eta^2\lambda\frac{1}{n_O}\sum_{i=1}^{n_O}\sigma_c'(f_i(\boldsymbol{w}))^2\left\|\boldsymbol{h}_i(\boldsymbol{o}_i,\boldsymbol{w}_t)\right\|^2, \tag{26}$$

*respectively.*

**Remark 15** (Robust Loss Kernel's Derivative $\sigma_c'$)**.** *We see here that the presence of the coefficient weight $\sigma_c'(f_i(\boldsymbol{w}))^2$ helps control the variance. Observe that $\sigma_c'(f_i(\boldsymbol{w}))$ tends to be small for outliers. If it is inversely proportional to $\|\boldsymbol{h}_i(\boldsymbol{o}_i,\boldsymbol{w})\|$ then the outlier variance can be greatly mitigated. We observe this phenomena in experiments. Figure 2 shows the impact for the case of linear regression. Note that this insight is missed when using the notion of noise-tolerant losses (see Section 2.2).*

In the next subsection, we will see how the same variance bound determines the region of convergence for each of the training algorithms.

### 6.3 Increased Region of Convergence

We now prove convergence results of the graduated non-convexity and adaptive training algorithms. We also derive convergence results for stochastic gradient descent as they serve as a good comparison. Note that our goal is to converge to the outlier-free optima $f_I^*$, and not the global optima of a robust estimation problem. We make two structural assumptions on the outlier-free objectives, *i.e.*, $f_{i,I}(\boldsymbol{w})$. We assume them to be $L$-smooth and $\mu$-Polyak-Lojasiewicz.

**Definition 16** ($L$-smooth)**.** *A continuously differentiable function $f$ is said to be $L$-smooth if it satisfies*

$$f(\boldsymbol{y}]) \leq f(\boldsymbol{x}) + \nabla f(\boldsymbol{x})^\mathsf{T}(\boldsymbol{y}-\boldsymbol{x}) + \frac{L}{2}\left\|\boldsymbol{y}-\boldsymbol{x}\right\|^2. \tag{27}$$

**Definition 17** ($\mu$-Polyak-Lojasiewicz)**.** *A continuously differentiable function $f$ is said to be $\mu$-Polyak-Lojasiewicz is*

$$f(\boldsymbol{w}) - \min_{\boldsymbol{w}} f(\boldsymbol{w}) \leq \frac{1}{2\mu}\left\|\nabla f(\boldsymbol{w})\right\|^2. \tag{28}$$

We remark that if $f_i$s are all $L$-smooth or $\mu$-Polyak-Lojasiewicz, then $f = \frac{1}{n}\sum_{i=1}^{n}f_i$ is also $L$-smooth and $\mu$-Polyak-Lojasiewicz (Garrigos & Gower, 2023).

Using this machinery, we first derive the region of convergence for the stochastic gradient descent algorithm solving (11).

**Theorem 18** (Convergence of Stochastic Gradient Descent)**.** *Let $f_{i,I}$ be $L$-smooth and $\mu$-Polyak-Lojasiewicz. Then, the stochastic gradient descent algorithm (with update $\boldsymbol{w}_{t+1} = \boldsymbol{w}_t - \eta\nabla f_i(\boldsymbol{w}_t)$) converges to the optimal value, namely $\mathbb{E}[\|f_I(\boldsymbol{w}_t) - f_I^*\| \,|\boldsymbol{w}_0] < \epsilon$, provided all the model weights $\boldsymbol{w}_t$ lie in the region $W_{SGD}$ given by*

$$W_{SGD} = \left\{\boldsymbol{w}\in\mathbb{R}^d \;\middle|\; \frac{1}{n_O}\sum_{i\in n_O}\left\|\boldsymbol{h}_i(\boldsymbol{o}_i,\boldsymbol{w})\right\|^2 < M\right\},$$

*and $\eta < \frac{\mu}{L}\min\left\{\frac{1}{L}, \frac{\epsilon}{3\lambda M + 2L\Delta_{f_I}}\right\}$, for some $M > 0$; where $\Delta_{f_I} = \frac{1}{n}(f_I^* - \min_{\boldsymbol{w}} f_{i,I}(\boldsymbol{w}))$.*

We next analyze the region of convergence of the adaptive algorithm.

**Theorem 19** (Convergence of Adaptive Algorithm)**.** *Let $f_{i,I}$ be $L$-smooth and $\mu$-Polyak-Lojasiewicz. Furthermore, let $\nabla f_{i,I}(\boldsymbol{w})^\mathsf{T}\nabla f_I(\boldsymbol{w}) \geq 0$ for all $i$. Then, the adaptive method (with update $\boldsymbol{w}_{t+1} = \boldsymbol{w}_t - \eta\sigma_{c_t}'(f_i(\boldsymbol{w}_t))\nabla f_i(\boldsymbol{w}_t)$ and $c_t$ chosen such that $\frac{1}{n}\sum_{i=1}^{n}\sigma_{c_t}'(f_i(\boldsymbol{w}_t)) = \zeta$) converges to the optimal value, namely $\mathbb{E}[\|f_I(\boldsymbol{w}_t) - f_I^*\| \,|\boldsymbol{w}_0] < \epsilon$, provided all the model weights $\boldsymbol{w}_t$ lie in the region $W_\rho$ given by*

$$W_\sigma = \left\{\boldsymbol{w}\in\mathbb{R}^d \;\middle|\; \begin{array}{l}\exists c \; s.t. \quad \frac{1}{n_O}\sum_{i\in n_O}\sigma_c'(f_i(\boldsymbol{w}))^2\left\|\boldsymbol{h}_i(\boldsymbol{o}_i,\boldsymbol{w})\right\|^2 < M, \\ \frac{1}{n}\sum_{i=1}^{n}\sigma_c'(f_i(\boldsymbol{w})) = \zeta, \; and \; \min_i\sigma_c(f_i(\boldsymbol{w})) \geq \phi > 0\end{array}\right\}, \tag{29}$$

*and $\eta < \frac{\mu\phi}{L}\min\left\{\frac{1}{L}, \frac{\epsilon}{3\lambda M + 2L\Delta_{f_I}\zeta}\right\}$, for some $M > 0$; where $\Delta_{f_I} = \frac{1}{n}(f_I^* - \min_{\boldsymbol{w}} f_{i,I}(\boldsymbol{w}))$.*

The set $\mathsf{W}_\sigma$ has two more constraints $\frac{1}{n}\sum_{i=1}^n \sigma_c'(f_i(\boldsymbol{w})) = \zeta$ and $\min_i \sigma_c(f_i(\boldsymbol{w})) \geq \phi > 0$. The first comes from the step to update the parameter $c$ in the algorithm and is always satisfied. The second is an assumption we make for the proof to hold. This will hold true for all continuously differentiable $\sigma$. Therefore, the key constraint that determines $\mathsf{W}_\sigma$ is

$$\frac{1}{n_O}\sum_{i \in n_O} \sigma_c'(f_i(\boldsymbol{w}))^2 \left\|\boldsymbol{h}_i(\boldsymbol{o}_i, \boldsymbol{w})\right\|^2 < M. \tag{30}$$

Comparing this to the constraint that defines $\mathsf{W}_{\text{SGD}}$ we see a multiplicative factor of $\sigma_c'(f_i(\boldsymbol{w}))^2$ appear before the summation.

**Remark 20** (Increased Region of Convergence). *Firstly, note that both the algorithms converge when there are no outliers. The presence of outliers shrinks the region of convergence for both the stochastic gradient descent and the adaptive method. However, the region of convergence for the adaptive method, i.e., $\mathsf{W}_\sigma$, is larger than $\mathsf{W}_{SGD}$. This is because the constraint (30) is weaker than the one that defines $\mathsf{W}_{SGD}$. Thus, thee use of the robust loss kernel $\sigma$ (and the coefficient weighting with $\sigma'$) widens the region of convergence.*

**Remark 21** (Convergence and the Fraction of Outliers $\lambda$). *Robust statistics literature has investigated the notion of breakdown point, which is a fraction of outlier samples that the estimator can handle, after which the estimator can produce arbitrarily bad estimates (Huber (1981)). A similar notion could be investigated for robust training algorithms. However, we have found it hard to obtain an explicit relation between convergence and the fraction of outliers $\lambda$ in the training data. We assume arbitrary outliers and a very general model of how the outliers interact with the model, i.e., via an additive gradient term: $\boldsymbol{h}_i(\boldsymbol{o}_i, \boldsymbol{w})$. We believe that it is not possible to obtain a clean relation between convergence and $\lambda$ in such a general case. For example, even a few, really bad outliers can affect convergence (see $\mathsf{W}_{SGD}$). Our result instead shows how the robust loss kernel $\sigma_c$ affects the impact of outliers $\boldsymbol{h}_i(\boldsymbol{o}_i, \boldsymbol{w})$ via $\sigma_c'$ towards determining the region of convergence (see $\mathsf{W}_\sigma$).*

The proof of Theorem 19 relies on deriving an iterative relation between $\delta_{t+1}$ and $\delta_t$, where $\delta_t = \mathbb{E}[f_I(\boldsymbol{w}_t) - f_I^*|\boldsymbol{w}_0]$. The problem that the graduated non-convexity algorithm poses is that the coefficient weights $u_{i,t}$ at iteration $t$ are determined by $\boldsymbol{w}_s$ at iteration $s$, for all $t \in [s, s+T]$ and $s \in \{0, T, 2T, \ldots\}$. The proof of Theorem 19 can be extended to obtain the following result for the graduated non-convexity algorithm.

**Theorem 22** (Convergence of Graduated Non-Convexity Algorithm). *Let $f_{i,I}$ be $L$-smooth and $\mu$-Polyak-Lojasiewicz. Furthermore, let $\nabla f_{i,I}(\boldsymbol{w})^\mathsf{T}\nabla f_I(\boldsymbol{w}) \geq 0$ and let $R(\boldsymbol{w})$ denote the region where all past $T$ iterates lie (i.e., $\boldsymbol{w}_{t'} \in R(\boldsymbol{w}_t)$ for all $t' \in [t-T, t]$), given $\boldsymbol{w}_t = \boldsymbol{w}$. Then, the graduated non-convexity algorithm (with update $\boldsymbol{w}_{t+1} = \boldsymbol{w}_t - \eta\sigma_{c_t}'(f_i(\boldsymbol{w}_s))\nabla f_i(\boldsymbol{w}_t)$, for all $t \in [s, T+s]$ and $s \in \{0, T, 2T, \ldots\}$, and $c_t$ chosen such that $\frac{1}{n}\sum_{i=1}^n \sigma_{c_t}'(f_i(\boldsymbol{w}_s)) = \zeta$ for all $s$) converges to the optimal value, namely $\mathbb{E}[\|f_I(\boldsymbol{w}_t) - f_I^*\| \,|\boldsymbol{w}_0] < \epsilon$, provided all the model weights $\boldsymbol{w}_t$ lie in the region $\mathsf{W}_{\sigma-GNC}$ given by*

$$\mathsf{W}_{\sigma-GNC} = \left\{\boldsymbol{w} \in \mathbb{R}^d \;\middle|\; \begin{array}{c} \max_{(\boldsymbol{w}',c) \in H(\boldsymbol{w})} \frac{1}{n_O}\sum_{i \in n_O} \sigma_c'(f_i(\boldsymbol{w}'))^2 \left\|\boldsymbol{h}_i(\boldsymbol{o}_i, \boldsymbol{w})\right\|^2 < M \\ and \;\; \min_i \sigma_c(f_i(\boldsymbol{w})) \geq \phi > 0 \end{array}\right\}, \tag{31}$$

*for some $M > 0$, where $H(\boldsymbol{w}) = \left\{(\boldsymbol{w}',c) \;\middle|\; \boldsymbol{w}' \in R(\boldsymbol{w}) \text{ and } c \text{ s.t. } \frac{1}{n_O}\sum_{i \in n_O} \sigma_c'(f_i(\boldsymbol{w}')) = \zeta\right\}$, provided $\eta < \frac{\mu\phi}{L}\min\left\{\frac{1}{L}, \frac{\epsilon}{3\lambda M + 2L\Delta_{f_I}\zeta}\right\}$ with $\Delta_{f_I} = \frac{1}{n}(f_I^* - \min_{\boldsymbol{w}} f_{i,I}(\boldsymbol{w}))$.*

Unlike the adaptive method, the graduated non-convexity algorithm requires the quantity

$$\frac{1}{n_O}\sum_{i \in n_O} \sigma_c'(f_i(\boldsymbol{w}'))^2 \left\|\boldsymbol{h}_i(\boldsymbol{o}_i, \boldsymbol{w})\right\|^2, \tag{32}$$

to remain bounded, where $\boldsymbol{w}'$ are the model weights of any of the previous $T$ iterations. The space $H(\boldsymbol{w})$ is going to be larger for larger $T$, which makes sense, as belief about the outliers computed $T$ iterations earlier is likely to be stale now and impact convergence.

**Remark 23** (Convergence in Robust Estimation). *A line of prior work has investigated convergence of iterative re-weighted least square type algorithms, proposed to mitigate the presence of outliers. Aftab &*

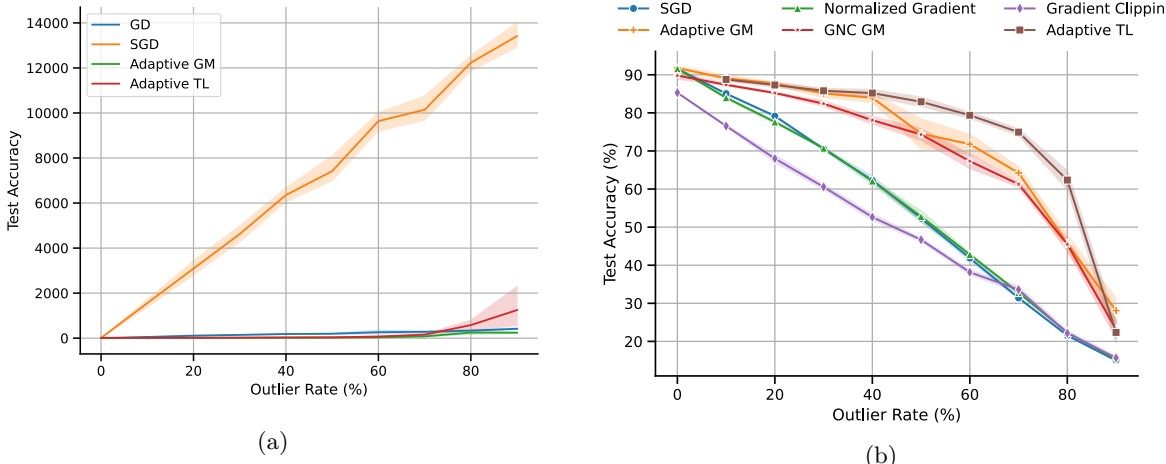

(a)                                               (b)

Figure 3: (a) Test accuracy (*i.e.,* RMSE on test data) as a function of outlier fraction $\lambda$ in the training data. Shows the gradient descent (GD) algorithm, stochastic gradient descent (SGD) algorithm, and two adaptive training algorithms with TL and GM robust losses. (b) Test classification accuracy as a function of outlier fraction $\lambda$ in the training data. Shows SGD (CE): stochastic gradient descent with cross-entropy loss, SGD (MAE): stochastic gradient descent with mean absolute error loss, Gradient clipping with MAE loss, GNC with GM loss, and two adaptive training algorithms with GM and TL loss.

*Hartley (2015) were the first to observe that it is the concavity property of $\rho(\sqrt{r})$ that ensures that the loss decreases for the iterative re-weighted least squares (here $\rho$ is the robust loss as in Section 2.1). They argued for concavity of $\rho(\sqrt{r})$ to be a necessary property for every robust loss design. This property translates to concavity of the robust loss kernel $\sigma$ and is satisfied by Definition 6. Recent work (Peng et al., 2023) derived two new graduated non-convexity algorithms for robust estimation, and for the first time, proved that they converge to the local optima of the robust M-estimation objective, albeit perturbed by $\epsilon$. These convergence results however did not investigate convergence of the iterates to the outlier-free optima $f_I^*$. We believe that developing convergence results to the outlier-free optima to be the correct objective to set.*

*These works focus on the robust estimation problems and, therefore, do not consider the deep learning setup where the training is inherently stochastic due to finite batch sizes.*

## 7 Experiments

We experimentally demonstrate the theoretical results. We show that the graduated non-convexity and adaptive training algorithms achieve lower variance in gradient computation and lead to better outlier mitigation. We observe that the methods are able to retain performance even when the percentage of outliers $\lambda$ is large. We demonstrate this in three applications: linear regression, image classification, and neural scene rendering (Mildenhall et al., 2020; Müller et al., 2022; Tancik et al., 2023). The first two experiments primarily show the general applicability of our training algorithms and validate the theoretical results. The third experiment shows that the algorithms can be applied to mitigate pixel-level outliers in novel view synthesis problem, that uses neural radiance fields.

We implement three methods: (i) *Adaptive TL*: adaptive training algorithm with truncated loss kernel, (ii) *Adaptive GM*: adaptive training algorithm with Geman McClure loss kernel, (iii) *GNC GM*: graduated non-convexity algorithm with Geman McClure loss kernel. See Table 1 for all the robust loss kernels.

### 7.1 Linear Regression

We first consider the simple problem of linear regression. Given $N$ measurement pairs $(\boldsymbol{x}_i, y_i) \in \mathbb{R}^k \times \mathbb{R}$, we estimate a vector $\hat{\boldsymbol{w}} \in \mathbb{R}^k$ that minimizes a mean squared error (MSE) loss $f(\boldsymbol{w}) = \frac{1}{n} \sum_{i=1}^{n} (y_i - \boldsymbol{w}^T \boldsymbol{x}_i)^2$. We generate the measurement pairs $(\boldsymbol{x}_i, y_i)$ by first sampling each coordinate of $\boldsymbol{x}_i$ uniformly randomly from $(0, 1]$ and $\boldsymbol{w}^\star$ from $\mathcal{N}(0, 1)$, and compute $y_i = \boldsymbol{w}^\star \boldsymbol{x}_i + \epsilon_i + o_i$, where $\epsilon_i \sim \mathcal{N}(0, 0.1)$ is a noise term and $o_i$

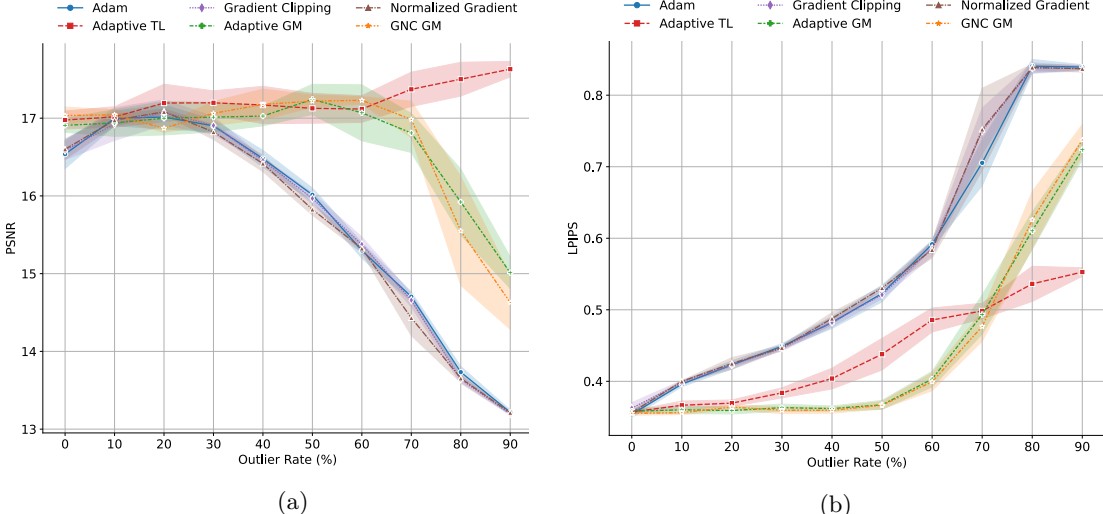

(a)                                                          (b)

Figure 4: Test accuracy of the trained model as a function of %outliers in the training data for various training algorithms: (i) Adam / SGD, the baseline approach proposed for training without outliers; (ii) Gradient Clipping, (iii) Normalized Gradient, (iv) Adaptive TL, (v) Adaptive GM, and (vi) GNC GM.

is the outlier term sampled from $\mathcal{N}(0, 5)$, if $i \in n_O$, and is otherwise set to zero. We vary $\lambda$ (the fraction of outliers, *i.e.,* $n_O/n$) from 0% to 90% with a 10% increment. We average over five Monte-Carlo trials for each $\lambda$. For all methods, step size $\eta$ is set to $7 \times 10^{-4}$ and number of iterations is fixed at $10^4$. We use batch size of one in training.

Figure 3a plots the test accuracy (*i.e.,* root mean squared error (RMSE)) as a function of fraction of outliers $\lambda$ in the training data. We observe that even though MSE is noise-tolerant (see Section 2.2), the SGD algorithm does not converge. This is because the outliers tend to induce high-variance during each descent iteration. Figure 2 shows a training instance and how the variance affects convergence. The Adaptive GM and Adaptive TL reduce this variance and show better convergence. Gradient descent converges to the outlier-free optima correctly. This shows that the notion of noise-tolerance is useful when you have low variance in the estimation of gradients.

## 7.2   Image Classification

We train a standard DLA-34 (Yu et al., 2018) network on the CIFAR10 datasets, with the standard train and test splits. All methods are trained with a total of 500 epochs and the batch size of 128 and use cross entropy loss. To generate noisy labels, we adopt the standard symmetric noise model where sample labels are replaced following a uniform distribution of probability. We vary the fraction of outliers $\lambda$ in the training set from 0% to 90% with 10% increment. We implement SGD with momentum with fixed learning rate of 1e-3 and a momentum of 0.9, and use it as the gradient-based training algorithm in the implementation of the graduated non-convexity and adaptive algorithms. For all methods a weight decay of 5e-4 is applied during training.

Figure 3b plots test accuracy as a function of the outlier ratio $\lambda$. We observe that the Adaptive TL, Adaptive GM, and GNC GM show improved mitigation of outliers as opposed to simply training with the SGD algorithm. This validates our results in Section 6.3 which argue that the graduated non-convexity and adaptive training algorithms have a larger region of convergence.

## 7.3   Neural Radiance Field

We employ the open-source Nerfacto (Tancik et al., 2023) model, a popular implicit scene reconstruction pipeline that combines Instant-NGP (Müller et al., 2022) with a camera-pose refine-

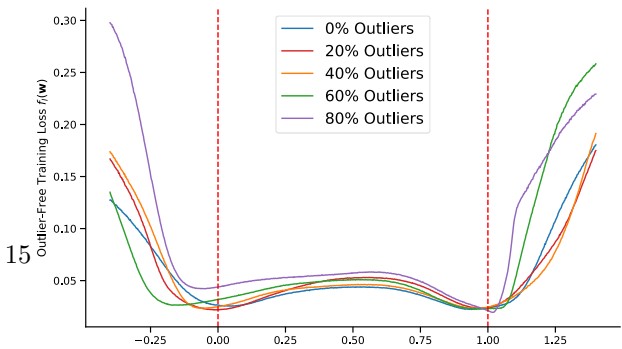

ment stage. We use the default model configuration parameters provided and an exponential decay scheduler with 2e5 steps with a final learning rate of 1e-4. We simulate pixel-level noise by adding uniformly distributed noise to the camera ray originating from each pixel, which is selected with probability $\lambda$. We use the Adam optimizer with a learning rate of 1e-3.

Figure 4 plots two test accuracy metrics as a function of outlier rate $\lambda$. We again observe that the Adaptive TL, Adaptive GM, and GNC GM show better robustness to outliers in the training data. Adaptive TL performs the best and shows good mitigation of outliers even when the training images have 90% of the pixels degraded with outliers. To investigate a little more deeply about where the Adam convergence vis-a-vis our algorithms, we plot the 1D loss landscape in Figure 5. It plots the loss landscape as a function of an interpolation parameter $\kappa$. The x-axis point 1 is the optimal model weight converged to by the Adaptive TL training, and the x-axis point 0 is the optimal model weight converged to by the vanilla Adam. We observe that the point to which Adam converges is a different local minima and is unstable in presence of outliers (*i.e.,* we see the loss landscape wobble as $\lambda$ changes). Whereas the loss landscape near the model weight that Adaptive TL converges to remains the same, across the outlier rate $\lambda$. Figure 6 shows views synthesized by two models: one trained with Adaptive TL and another trained with vanilla Adam, when we have an outlier rate of 80% during training. We observe that the vanilla Adam is not able to recover any reasonable visual signal after training, while the Adaptive TL sees a visually good view synthesis.

## 8 Related Work

### 8.1 Outlier Robust Training of Deep Learning Models

Training image classification models, in presence of outliers in the training data, has been well investigated in the last decade. It has been treated as a good proxy task for developing outlier robust training algorithms for training deep learning models. Several methods have been proposed to mitigate the presence of outliers in the training data. These include label correction methods, loss correction methods, refined training strategy, and robust loss function design. Algan & Ulusoy (2021); Song et al. (2023) provide a detailed review on the topic of training multi-label classifiers in presence of outliers in the training data. The state-of-the-art approaches (*e.g.,* Li et al. (2020)) always use a combination of these approaches to attain the best results. While most approaches remain specific to the task of image classification, some of them are generally applicable. Two such approaches include robust loss design and outlier robust training algorithms.

Seminal works Ghosh et al. (2015; 2017) introduce the notion of *noise-tolerant loss* (if $\boldsymbol{w}_\lambda^*$ denotes the optimal model weights when minimizing a loss function $l$, then $l$ is said to be noise-tolerant to $\lambda$ fraction of outliers if $\boldsymbol{w}_\lambda^* = \boldsymbol{w}_0^*$). The paper goes on to prove that symmetric losses, such as a simple mean absolute error (MAE), is noise tolerant for multi-label classification provided $\lambda < 1 - 1/K$, where $K$ denotes the set of all label classes. The classical cross entropy (CE) loss is shown to be not noise tolerant. Several works since then have investigated the design of robust and noise-tolerant losses. Zhang & Sabuncu (2018) propose generalized cross entropy (GCE) loss that generalizes MAE and the CE loss, and is inspired by the negative Box-Cox transformation (Box & Cox, 1964) and the generalized maximum likelihood framework (Ferrari & Yang, 2010). Amid et al. (2019) replace the logarithms and exponentials in the cross-entropy loss with 'tempered' versions (Naudts, 2002). The temperature parameters are tuned to achieve better outlier robustness. Wang et al. (2019) propose symmetric cross entropy loss, along the lines of symmetric KL divergence. Feng et al. (2020) propose a loss that is a finite Taylor series expansion of the log likelihood loss. Zhou et al. (2023) propose an asymmetric loss and show how popular robust losses can be turned into noise-tolerant losses, under dominant clean label assumption (*i.e.,*, the label noise is such that clean label remains dominant in the noise induced distribution). Ma et al. (2020) show that any loss can be converted to noise-tolerant loss by applying a simple normalization. However, this changes the structure of the loss and can cause underfitting or non-convergence. They propose active-passive loss that combines two noise-tolerant loss functions that

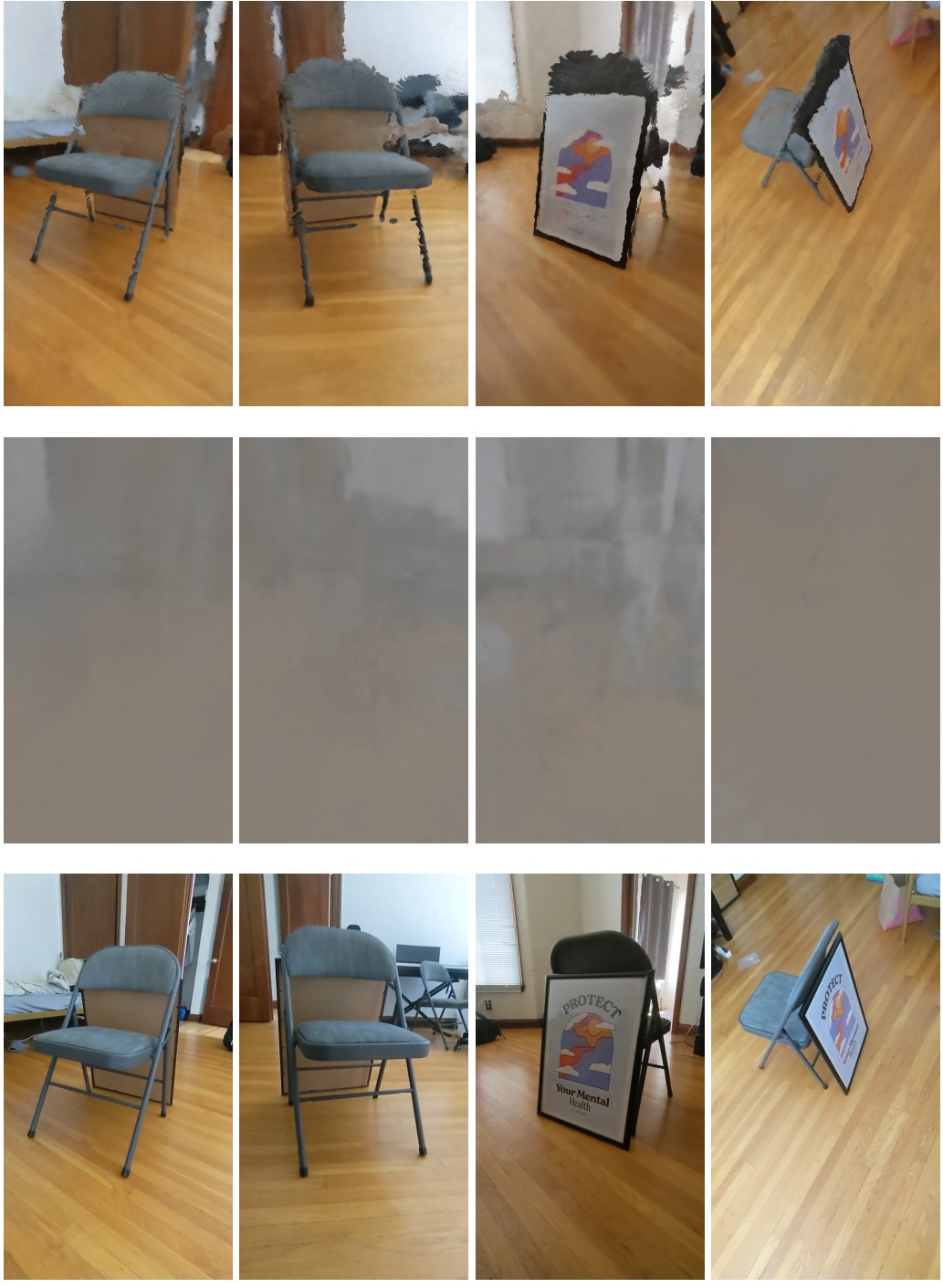

Figure 6: Nerfacto reconstruction results after 80% of the training pixels have been perturbed by noise. The first row shows the result of training with our Adaptive Training Algorithm with Truncated Loss. The second row shows the result of running the originally proposed Adam. The third row shows the ground truth images from the corresponding views.

can boost each other. Curriculum and peer losses are proposes in Lyu & Tsang (2020) and Liu & Guo (2020), respectively. Xu et al. (2019) propose determinant-based mutual information loss and show that it can successfully tackle instance-independent noise.

Outlier robustness is not only achieved by designing better robust losses, but also by developing better training strategies. Zhang et al. (2018) propose using convex combination of training samples to have the networks favor linear behaviors. Elesedy & Hutter (2023) maintain a buffer of clipped gradients and add them to the next iteration; shows that their clipped updates are unbiased; convergence guaranteed under some assumptions. DivideMix Li et al. (2020) propose a semi-supervised approach to refine noisy labels during training using a mixture model. Menon et al. (2020) investigate the effect of gradient clipping on countering label noise in training classification networks. Mai & Johansson (2021) develop quantitative results on the convergence of clipped stochastic gradient descent for non-smooth convex functions. Ren et al. (2018) propose an iterative re-weighting scheme in training machine learning models in presence of outliers. It uses a small set of clean samples to evaluate and update the weights at each iteration. Recent works have considered training neural radiance fields (NeRF) in presence of distractors (*e.g.,* moving objects, lighting variations, shadows). RobustNeRF by Sabour et al. (2023), among other heuristics, uses median to trim outliers in loss computation during training. *While these training methods have shown promising results, their convergence has not been analyzed.*

## 8.2 Robust Estimation in Robot and Computer Vision

Outlier-infested data is common in robot and computer vision. Examples include camera pose estimation, point cloud registration, relative camera pose estimation, fundamental matrix estimation, and robot state estimation via pose graph optimization. It is very often the case that one needs to segregate the outlier measurements, while simultaneously fitting the model to the remaining measurements. Three frameworks dominate the solution approaches. The first is to formulate the problem as a robust estimation problem (*e.g.,* M-estimator (Huber, 1981)), and then solve it by reframing it as a polynomial optimization problem (*e.g.,* Yang et al. (2020b); Yang & Carlone (2022)). While these methods yield globally optimal solution, with certification guarantees, it tends to be expensive in the compute-time requirements. The second framework is based on RANSAC (Fischler & Bolles, 1981). In it, you choose a small subset of measurements, and solve the problem using only those measurements (*i.e.,* using minimal solvers). Then, identify all the other measurements that are consistent with the solution to determine when to stop. RANSAC is a time-tested method, and can be fast, but is not guaranteed to converge to the globally optimal solution.

Graduated Non-Convexity (GNC) (Antonante et al., 2021; Yang et al., 2020a; Black & Rangarajan, 1996; Peng et al., 2023) have emerged as a good balance between real-time computation and effective outlier mitigation for state estimation problems in robotics. In it, the robust M-estimation problem is re-framed as an iterative re-weighted least squares. The weights indicates whether a measurement is an inlier or an outlier. The duality result established in Black & Rangarajan (1996) enables one to do this. This is popularly known as the Black-Rangarajan duality and is a common technique used to re-formulate and solve robust estimation problems. Iterative re-weighted (non-linear) least squares updates the weights in each iteration.

While GNC has shown promising results, a theoretical understanding about its convergence has been lacking in the robust estimation literature. Aftab & Hartley (2015) show that an iterative re-weighted least squares (IRLS) scheme, where the weights are updated according to the Black-Rangarajan duality, reduces the M-estimator loss, and can attain optimality, under very strict convexity conditions (*i.e.,* it requires the objective in (1) to be convex). The IRLS scheme has been particularly studied in solving the Fermat-Weber problem. In it, the goal is to find a point that minimizes the $l_p$ distance from a given set of points. Brimberg & Love (1993) investigates convergence of the IRLS procedure in this setting, whereas Aftab et al. (2015) extends the IRLS scheme over Riemanian manifolds (*e.g.,* SO(3)) and proves convergence for rotation averaging problems. These works primarily tackle the case where the robust loss is assumed to be fixed.

Recent works have proposed parameterized (aka adaptive) robust losses to enable automatic tuning. Tavish & Barfoot (2015) was the earliest work in robot state estimation to show that adaptive robust losses improve outlier rejection. Barron (2019) develops a general and adaptive robust loss function, that instantiates other well known robust losses for different choices of the adaptive parameter. Chebrolu et al. (2021) uses this

general robust loss and adapts its shape in training to better mitigate the outliers in robot state estimation problems. The GNC algorithm by Yang et al. (2020a); Antonante et al. (2021) has shown good practical performance, however, does not have any theoretical convergence guarantees. Peng et al. (2023) propose new GNC algorithms, by defining two new parameterized versions of the robust losses for $l_p$ and the truncated least squares loss. Unlike in Yang et al. (2020a), they prove that their GNC algorithm converges to stationary points of the M-estimator, albeit perturbed by $\epsilon$. Shen & Sanghavi (2019) proposes iteratively training with a pre-defined fraction of 'good' samples (*i.e.,* samples with the lowest loss). This algorithm can be thought of as using an adaptive truncated robust loss in each iteration, where the truncation threshold is adapted at each iteration. The paper also derives convergence to error bounds for a generalized linear model. *While there is interest in developing better GNC algorithms with convergence properties, these results do not directly extend to the context of training deep learning models. This is because the training procedure for deep learning models is different (*e.g., *finite batch sizes) than solving an optimization problem.*

## 8.3 Convergence Analysis of Training Algorithms in Presence of Outliers

Classical machine learning problems (*e.g.,* linear regression, principle component analysis, matrix decomposition) in presence of outliers have received significant attention, and many algorithms have been proposed that are robust. Training deep learning models in presence of outliers, however, is challenging for at least two reasons. First, deep learning models are trained using batches of data, and therefore, any algorithm only has access to an estimate of the true gradient. A biased or an outlier gradient can significantly affect convergence. Second, the training loss can be non-convex and is hard to analyze without making certain assumptions.

Stochastic gradient descent is the most popular approach for training deep learning models. Several works have investigated its convergence behavior, brought to forth its limitations (*e.g.,* noise-variance issue, biased gradient estimates, convergence to non-flat local optima), and proposed variants to overcome them (see Demidovich et al. (2023); Zhang et al. (2020a;b); Reisizadeh et al. (2023); Koloskova et al. (2023); Gower et al. (2020); Foret et al. (2020)). Garrigos & Gower (2023) provides a comprehensive review on analysis techniques for proving convergence of the SGD algorithms, under different assumptions on the training loss such as $L$-smoothness, strong convexity, and $\mu$-Polyak-Lojasiewicz. While analyzing SGD and its variants have been easier, convergence of the popular Adam optimizer remains elusive (Dereich & Jentzen, 2024).

Very few works have considered the effect of outliers on the convergence of training algorithms, including SGD. Menon et al. (2020) were the first to point out that gradient clipping (albeit small modification) can be robust to outliers. They analyzed the special case of linear classification, with training batch size of one, and showed its equivalence to minimizing a Huberized and partially Huberized losses. They showed that their proposed gradient clipping algorithm provably exhibits a constant excess risk under symmetric label noise, in binary classification. Merad & Gaïffas (2024) proposes gradient quantile clipping, where the gradient clipping threshold is chosen to be the $p$th quantile of all the estimated gradient norms. The paper goes on to derive convergence property of the iterates, under $L$-smoothness and strong convexity assumptions. Chhabra et al. (2024) draws a connection between identifying detrimental training sample (*i.e.,* a training sample that can unduly affect the model) and outlier gradient detection. Hu et al. (2024) formulates an adversarial training process, where for each given input-output sample, one estimates a worst-case input for each annotated output, and trains using the worst-case input cum output pair. The paper analyzes its $\mathcal{H}$-consistency, generalizability, and convergence for the special case of binary classification. Shen & Sanghavi (2019); Shah et al. (2020) proposes to iteratively train the model with a subset of samples that have the lowest loss. It shows convergence results under strong convexity and bounded variance of gradient estimates used in the stochastic gradient descent. Prasad et al. (2020) proposes to robustly estimate the gradients, and shows convergence under two outlier models on gradients, namely, Huber contamination and heavy-tail distribution. The analysis in all these papers is stochastic in nature, *i.e.,* they assume an outlier distribution. *On the contrary, our work studies the convergence of training algorithms in presence of arbitrary outliers, without any distributional assumption.*

# 9 Conclusion

The unified robust loss kernel $\sigma$ creates an opportunity to cross-pollinate, *i.e.,* test robust kernels developed in the deep learning literature in robust estimation problems, and vice versa. The modified Black-Rangarajan duality can now be applied to any machine learning problem, and not just those that adhere to a least squares loss. The two class of robust training algorithms are generic tools in the practitioner's arsenal to robustly train machine learning models, in presence of outliers. The analysis technique developed opens the doors to further study convergence of training algorithms, under arbitrary outliers assumption. While we present a general result, specific problem structure may be exploited in special cases, in the future, to derive the impact of robust loss kernels on convergence. The exact impact of the parameter update rule and tighter convergence bound for the graduated non-convexity algorithm remain open.

State-of-the-art results tend to be obtained by combining a mix of techniques (Li et al., 2020). We hope that the researchers find the robust training algorithms useful towards achieving state-of-the-art result in their specific task.

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

## A  Proof of Corollary 4

**Derivation from First Principles.** We show that the following two optimization problems are equivalent:

$$\underset{r}{\text{Minimize}} \; \sigma(r) \quad \equiv \quad \underset{r,u}{\text{Minimize}} \; u \cdot r + \Phi(r), \tag{33}$$

if $\Phi(u) = \sigma((\sigma')^{-1}(u)) - u(\sigma')^{-1}(u)$. This establishes the core of modified Black-Rangarajan duality. Applying this to a sum of losses directly yields Corollary 4.

Let $u(r)$ be the $u$ that minimizes $w \cdot r + \Phi(r)$. The first order condition suggests that $u(r)$ must satisfy

$$r + \Phi'(u(r)) = 0. \tag{34}$$

The equivalence (33) will hold if $\sigma(r) = r \cdot u(r) + \Phi(u(r))$. Taking derivative with respect to $r$ on both sides of this equation yields

$$\sigma'(r) = u(r) + ru'(r) + \Phi'(u(r))u'(r) = u(r), \tag{35}$$

where the last equality followed by using (34). Using (35) in $\sigma(r) = r \cdot u(r) + \Phi(u(r))$ we obtain

$$\sigma(r) = r \cdot \sigma'(r) + \Phi(\sigma'(r)). \tag{36}$$

Now, note that $\sigma$ is a robust loss kernel (Definition 6), and therefore satisfies $\sigma'(r) \in [0,1]$ and $\sigma''(r) < 0$, *i.e.*, $\sigma'$ is strictly monotonic and has an inverse. Therefore, let $u = \sigma'(r) \in [0,1]$ and $r = (\sigma')^{-1}(u)$. Substituting $r = (\sigma')^{-1}(u)$ in (36) yields

$$\Phi(u) = \sigma((\sigma')^{-1}(u)) - u(\sigma')^{-1}(u), \tag{37}$$

for $u \in [0,1]$. This proves the result.

## B  Robust Loss Kernels and Robust Losses

The first six robust loss kernels are given by $\sigma_c(r) = \text{cost.} \cdot \rho_{\sqrt{c}}(\sqrt{r})$, where $\rho_c(r)$ are the robust losses in Section 8.2; we have explicitly added the parameter $c$ in the notation $\rho_c(r)$. The constant multiple ensures that the kernel is scaled appropriately to satisfy Definition 6. The remaining robust loss kernels are obtained by substituting $r = -\log(\boldsymbol{p}[y])$ in the losses given in Section 8.1. This is because we want the losses (in Section 8.1) to be robust loss kernel of the cross-entropy loss. It can be analytically verified that all the kernels in Table 1 satisfy Definition 6.

## C Proof of Lemma 7

Setting the derivative of the objective to zero, we obtain

$$f_i(\boldsymbol{w}) = -\Phi'_{\sigma_c}(u^*). \tag{38}$$

We know from the modified Black-Rangarajan duality (Corollary 4) that $\Phi_{\sigma_c}(u) = -u(\sigma'_c)^{-1}(u) + \sigma_c((\sigma'_c)^{-1}(u))$. Taking its derivative we obtain

$$\Phi'_{\sigma_c}(u) = -(\sigma'_c)^{-1}(u). \tag{39}$$

Substituting this back in (38) and applying $\sigma'_c$ on both sides, we obtain the result.

## D Verifying Assumption 11

We first verify that the outlier gradient assumption (Assumption 11) holds for two broad class of problems, namely, non-linear regression and multi-label classification.

**Example 24** (Non-Linear Regression). *Consider a model $\boldsymbol{g}$ that predicts the output $\boldsymbol{y} = \boldsymbol{g}(\boldsymbol{w}, \boldsymbol{x})$ given the input $\boldsymbol{x}$ and model weights $\boldsymbol{w}$. The ith measurement loss is the L2 norm given by $f_i(\boldsymbol{w}) = \|\boldsymbol{y}_i - \boldsymbol{g}(\boldsymbol{w}, \boldsymbol{x}_i)\|^2$. The model is trained on annotated data, which suffers from incorrect output annotations: $\boldsymbol{y}_i = \boldsymbol{y}_i^*$ for $i \in n_I$, but $\boldsymbol{y}_i = \boldsymbol{y}_i^* + \boldsymbol{o}_i$ for $i \in n_O$; here $\boldsymbol{y}_i^*$ denotes the correct annotation. The loss for the outlier-infested measurement becomes $f_i(\boldsymbol{w}) = f_{i,I}(\boldsymbol{w}) + \|\boldsymbol{o}_i\|^2 + 2\boldsymbol{o}_i^\mathsf{T}(\boldsymbol{y}_i^* - \boldsymbol{g}(\boldsymbol{w}, \boldsymbol{x}_i))$, where $f_{i,I}(\boldsymbol{w}) = \|\boldsymbol{y}_i^* - \boldsymbol{g}(\boldsymbol{w}, \boldsymbol{x}_i)\|^2$. The gradient $\nabla f_i(\boldsymbol{w})$ of the outlier-infested objective is then given by*

$$\nabla f_i(\boldsymbol{w}) = \nabla f_{i,I}(\boldsymbol{w}) + \boldsymbol{h}_i(\boldsymbol{o}_i, \boldsymbol{w}), \tag{40}$$

*with $\boldsymbol{h}_i(\boldsymbol{o}_i, \boldsymbol{w}) = \nabla_{\boldsymbol{w}} g(\boldsymbol{w}, \boldsymbol{x}_i)\boldsymbol{o}_i$. This satisfies Assumption 11.*

**Example 25** (Multi-Label Classification). *Consider a model $\boldsymbol{p}(\boldsymbol{w}, \boldsymbol{x})$ that predicts the probability that the input $\boldsymbol{x}$ belongs to which class, i.e., $\boldsymbol{p}(\boldsymbol{w}, \boldsymbol{x})[y]$ denotes the predicted probability that the input $\boldsymbol{x}$ is of class $y \in [K]$. The model is trained on annotated data $\{(\boldsymbol{x}, y_i)\}_{i \in [n]}$. The loss component is given by*

$$f_i(\boldsymbol{w}) = -\log\left(\boldsymbol{p}(\boldsymbol{w}, \boldsymbol{x}_i)[y_i]\right). \tag{41}$$

*Annotations are not perfect: $y_i = y_i^*$ for $i \in n_I$, but this is not the case for outlier measurements. For outlier measurements, the loss can be re-written as*

$$f_i(\boldsymbol{w}) = -\log\left(\boldsymbol{p}(\boldsymbol{w}, \boldsymbol{x}_i)[y_i]\right), \tag{42}$$
$$= -\log\left(\boldsymbol{p}(\boldsymbol{w}, \boldsymbol{x}_i)[y_i^*]\right) - \log\left(\boldsymbol{p}(\boldsymbol{w}, \boldsymbol{x}_i)[y_i]/\boldsymbol{p}(\boldsymbol{w}, \boldsymbol{x}_i)[y_i^*]\right), \tag{43}$$
$$= f_{i,I}(\boldsymbol{w}) - \log\left(\boldsymbol{p}(\boldsymbol{w}, \boldsymbol{x}_i)[y_i]/\boldsymbol{p}(\boldsymbol{w}, \boldsymbol{x}_i)[y_i^*]\right). \tag{44}$$

*Therefore, $\nabla f_i(\boldsymbol{w}) = \nabla f_{i,I}(\boldsymbol{w}) + \boldsymbol{h}_i(\boldsymbol{o}_i, \boldsymbol{w})$, where*

$$\boldsymbol{h}_i(\boldsymbol{o}_i, \boldsymbol{w}) = \nabla_{\boldsymbol{w}}\left[-\log\left(\boldsymbol{p}(\boldsymbol{w}, \boldsymbol{x}_i)[y_i]/\boldsymbol{p}(\boldsymbol{w}, \boldsymbol{x}_i)[y_i^*]\right)\right], \tag{45}$$

*which satisfies Assumption 11.*

## E Proof of Lemma 14

For a batch size of one, the stochastic gradient descent and the adaptive algorithms the gradients are given by $\boldsymbol{g}_t = \eta \nabla f_i(\boldsymbol{w}_t)$ and $\boldsymbol{g}_t = \eta \sigma'_c(\boldsymbol{w}_t) \nabla f_i(\boldsymbol{w}_t)$, respectively, where $i$ is a uniformly distributed random variable over the set $[n]$, i.e., $i \sim \mathcal{U}([n])$. For stochastic gradient descent, note that the mean $\bar{\boldsymbol{g}}_t = \mathbb{E}_i[\eta \nabla f_i(\boldsymbol{w}_t)] =$

$\eta \nabla f_I(\boldsymbol{w}_t)$ because of the zero-mean assumption, *i.e.*, $\mathbb{E}_i[\boldsymbol{h}_i(\boldsymbol{o}_i, \boldsymbol{w})] = 0$. The variance is, therefore, given by

$$\mathbb{E}_i[\|\boldsymbol{g}_t - \eta \nabla f_I(\boldsymbol{w}_t)\|^2] = \frac{1}{n} \sum_{i=1}^{n} \|\eta \nabla f_{i,I}(\boldsymbol{w}_t) + \eta \boldsymbol{h}_i(\boldsymbol{o}_i, \boldsymbol{w}_t) - \eta \nabla f_I(\boldsymbol{w}_t)\|^2 \tag{46}$$

$$= \eta^2 \mathbb{E}_i[\|\nabla f_{i,I}(\boldsymbol{w}_t)\|^2] + \eta^2 \lambda \frac{1}{n_O} \sum_{i=1}^{n_O} \|\boldsymbol{h}_i(\boldsymbol{o}_i, \boldsymbol{w}_t)\|^2 - \eta^2 \|\nabla f_I(\boldsymbol{w}_t)\|^2$$

$$- 2\eta^2 \lambda \left( \frac{1}{n_O} \sum_{i=1}^{n_O} \boldsymbol{h}_i(\boldsymbol{o}_i, \boldsymbol{w}_t) \right)^\mathsf{T} \nabla f_I(\boldsymbol{w}_t) + 2\eta^2 \lambda \frac{1}{n_O} \sum_{i=1}^{n_O} \nabla f_{i,I}(\boldsymbol{w}_t)^\mathsf{T} \boldsymbol{h}_i(\boldsymbol{o}_i, \boldsymbol{w}_t).$$

Using the facts: (i) $\boldsymbol{h}_i$s are zero mean, (ii) $\mathbb{E}_i[\|\nabla f_{i,I}(\boldsymbol{w}_t)\|^2] \geq \|\nabla f_I(\boldsymbol{w}_t)\|^2$ due to Jensen's inequality, (iii) CauchyâĂŞSchwarz inequality along with Assumption 13, we obtain

$$\mathbb{E}_i[\|\boldsymbol{g}_t - \eta \nabla f_I(\boldsymbol{w}_t)\|^2] \leq 3\eta^2 \lambda \frac{1}{n_O} \sum_{i=1}^{n_O} \|\boldsymbol{h}_i(\boldsymbol{o}_i, \boldsymbol{w}_t)\|^2. \tag{47}$$

Following the same line of argument one can derive the result for the adaptive training algorithm.

## F    Preliminary Lemmas

We state and prove some results needed to establish the key result in the paper. We consider $i$ to be a uniformly distributed random variable over the set $[n] = \{1, 2, \ldots n\}$. The notation $\mathbb{E}_i[]$ refers to expectation with respect to $i$ and evaluates to a simple average: $\mathbb{E}_i[g(i)] = \frac{1}{n} \sum_{i=1}^{n} g(i)$.

**Lemma 26.** $\mathbb{E}_i [\nabla f_i(\boldsymbol{w})] = \nabla f_I(\boldsymbol{w}) + \lambda \frac{1}{n_O} \sum_{i \in n_O} \boldsymbol{h}_i(\boldsymbol{o}_i, \boldsymbol{w})$.

**Proof.** We know that $\nabla f_i(\boldsymbol{w}) = \nabla f_{i,I}(\boldsymbol{w}) + \boldsymbol{h}_i(\boldsymbol{o}_i, \boldsymbol{w})$ for all outlier measurements. For inlier measurements, $\nabla f_i(\boldsymbol{w}) = \nabla f_{i,I}(\boldsymbol{w})$ as $f_i(\boldsymbol{w}) = f_{i,I}(\boldsymbol{w})$. This implies

$$\mathbb{E}_i [\nabla f_i(\boldsymbol{w})] = \frac{1}{n} \sum_{i=1}^{n} \nabla f_i(\boldsymbol{w}), \tag{48}$$

$$= \frac{1}{n} \sum_{i \in n_I} \nabla f_{i,I}(\boldsymbol{w}) + \frac{1}{n} \sum_{i \in n_O} \nabla f_{i,I}(\boldsymbol{w}) + \boldsymbol{h}_i(\boldsymbol{o}_i, \boldsymbol{w}), \tag{49}$$

$$= \frac{1}{n} \sum_{i=1}^{n} \nabla f_{i,I}(\boldsymbol{w}) + \lambda \frac{1}{n_O} \sum_{i \in n_O} \boldsymbol{h}_i(\boldsymbol{o}_i, \boldsymbol{w}), \tag{50}$$

which proves the result. $\qquad\square$

**Lemma 27.** *Let the low singal-to-outlier ratio assumption (Assumption 13) hold. The second-order moments* $\mathbb{E}_i \left[ \|\nabla f_i(\boldsymbol{w})\|^2 \right]$ *and* $\mathbb{E}_i \left[ \nabla f_i(\boldsymbol{w})^\mathsf{T} \nabla f_I(\boldsymbol{w}) \right]$ *are bounded by:*

$$\mathbb{E}_i \left[ \|\nabla f_i(\boldsymbol{w})\|^2 \right] \leq 2L(f_I(\boldsymbol{w}) - f_I^*) + 2L\Delta_{f_I} + 3\lambda \frac{1}{n_O} \sum_{i \in n_O} \|\boldsymbol{h}_i(\boldsymbol{o}_i, \boldsymbol{w})\|^2, \tag{51}$$

*and*

$$\mathbb{E}_i \left[ \nabla f_i(\boldsymbol{w})^\mathsf{T} \nabla f_I(\boldsymbol{w}) \right] \geq 2\mu(f_I(\boldsymbol{w}) - f^*), \tag{52}$$

*where* $\Delta_{f_I} = \frac{1}{n} \sum_{i=1}^{n} (f_I^* - f_{i,I}^*)$ *and* $f_{i,I}^* = \min_{\boldsymbol{w}} f_{i,I}(\boldsymbol{w})$.

**Proof.** Expand $\mathbb{E}_i[\|\nabla f_i(\boldsymbol{w})\|^2]$ as

$$\mathbb{E}_i[\|\nabla f_i(\boldsymbol{w})\|^2] = \frac{1}{n}\sum_{i=1}^n \|\nabla f_i(\boldsymbol{w})\|^2, \tag{53}$$

$$= \frac{1}{n}\sum_{i\in n_I} \|\nabla f_{i,I}(\boldsymbol{w})\|^2 + \frac{1}{n}\sum_{i\in n_O} \|\nabla f_{i,I}(\boldsymbol{w}) + \boldsymbol{h}_i(\boldsymbol{o}_i, \boldsymbol{w})\|^2, \tag{54}$$

$$= \frac{1}{n}\sum_{i\in n_I} \|\nabla f_{i,I}(\boldsymbol{w})\|^2 \tag{55}$$

$$+ \frac{1}{n}\sum_{i\in n_O} \|\nabla f_{i,I}(\boldsymbol{w})\|^2 + \|\boldsymbol{h}_i(\boldsymbol{o}_i, \boldsymbol{w})\|^2 + 2\boldsymbol{h}_i(\boldsymbol{o}_i, \boldsymbol{w})^\mathsf{T}\nabla f_{i,I}(\boldsymbol{w}),$$

$$\leq \frac{1}{n}\sum_{i\in n_I} \|\nabla f_{i,I}(\boldsymbol{w})\|^2 + \frac{1}{n}\sum_{i\in n_O} \|\nabla f_{i,I}(\boldsymbol{w})\|^2 + 3\|\boldsymbol{h}_i(\boldsymbol{o}_i, \boldsymbol{w})\|^2, \tag{56}$$

$$= \frac{1}{n}\sum_{i=1}^n \|\nabla f_{i,I}(\boldsymbol{w})\|^2 + 3\lambda\frac{1}{n_O}\sum_{i\in n_O} \|\boldsymbol{h}_i(\boldsymbol{o}_i, \boldsymbol{w})\|^2, \tag{57}$$

where, in order to obtain (56), we use the Cauchy–Schwarz inequality and Assumption 13. Since $f_{i,I}$ is $L$-smooth we have $\|\nabla f_{i,I}(\boldsymbol{w})\|^2 \leq 2L(f_{i,I}(\boldsymbol{w}) - f_{i,I}^*)$. Using this fact and re-arranging terms in (57) by adding and subtracting $f^*$ we obtain (51).

Use Lemma 26 and expand $\mathbb{E}_i\left[\nabla f_i(\boldsymbol{w})^\mathsf{T}\nabla f_I(\boldsymbol{w})\right]$ as follows:

$$\mathbb{E}_i\left[\nabla f_i(\boldsymbol{w})^\mathsf{T}\nabla f_I(\boldsymbol{w})\right] = \|\nabla f_I(\boldsymbol{w})\|^2 + \lambda\frac{1}{n_O}\sum_{i\in n_O} \boldsymbol{h}_i(\boldsymbol{o}_i, \boldsymbol{w})^\mathsf{T}\nabla f_I(\boldsymbol{w}), \tag{58}$$

$$\geq 2\mu(f_I(\boldsymbol{w}) - f_I^*) - \lambda\frac{1}{n_O}\sum_{i\in n_O} \|\boldsymbol{h}_i(\boldsymbol{o}_i, \boldsymbol{w})\|\|\nabla f_I(\boldsymbol{w})\|, \tag{59}$$

where (i) $\|\nabla f_I(\boldsymbol{w})\|^2 \geq 2\mu(f_I(\boldsymbol{w}) - f_I^*)$ follows because $f_I$ is $\mu$-PL (as each of the $f_{i,I}$ are also $\mu$-PL), and (ii) the second part follows from Cauchy–Schwarz inequality. This implies

$$\mathbb{E}_i\left[\nabla f_i(\boldsymbol{w})^\mathsf{T}\nabla f_I(\boldsymbol{w})\right] \geq 2\mu(f_I(\boldsymbol{w}) - f_I^*).$$

$\square$

**Lemma 28.** *Let the low singal-to-outlier ratio assumption (Assumption 13) hold. Also, assume that $\frac{1}{n}\sum_{i=1}^n \sigma_c'(f_i(\boldsymbol{w})) = \zeta$. Furthermore, let $\nabla f_{i,I}(\boldsymbol{w})^\mathsf{T}\nabla f_I(\boldsymbol{w}) \geq 0$ and $0 < \phi \leq \inf_{\boldsymbol{w}\in\mathcal{W}}\sigma'(f_i(\boldsymbol{w}))$ then*

$$\mathbb{E}_i\left[\sigma_c'(f_i(\boldsymbol{w}))^2\|\nabla f_i(\boldsymbol{w})\|^2\right] \leq 2L(f_I(\boldsymbol{w}) - f^*) + 2Lf_I^*\zeta + 3\lambda\frac{1}{n_O}\sum_{i\in n_O} \sigma_c'(f_i(\boldsymbol{w}))^2\|\boldsymbol{h}_i(\boldsymbol{o}_i, \boldsymbol{w})\|^2,$$

*and*

$$\mathbb{E}_i\left[\sigma_c'(f_i(\boldsymbol{w}))\nabla f_i(\boldsymbol{w})^\mathsf{T}\nabla f_I(\boldsymbol{w})\right] \geq 2\phi\mu(f_I(\boldsymbol{w}) - f_I^*)$$

**Proof.** The proof follows the same line of argument as the proof of Lemma 27. $\square$

**Lemma 29.** *Consider the recurrence relation $\delta_{t+1} \leq (1 - a\eta)\delta_t + \eta f(\eta)c$, where $a$, $\eta$, and $c$ are positive constants, and $f(\eta)$ is some known scalar function of $\eta$. We have $\delta_T < \epsilon$ provided*

$$T > \frac{1}{a\eta}\log(2\delta_0/\epsilon), \quad f(\eta)c/a < \epsilon/2, \quad and \quad a\eta < 1, \tag{60}$$

*for all $\epsilon > 0$.*

**Proof.** Let $a\eta < 1$. Then composition the iterates (*i.e.*, $\delta_{t+1} \le (1-a\eta)\delta_t + \eta f(\eta)c$) from $t = 0$ to $t = T$ we obtain

$$\delta_T \le (1-a\eta)^T \delta_0 + \eta f(\eta)c \sum_{t=0}^{T-1} (1-a\eta)^t, \tag{61}$$

$$\le (1-a\eta)^T \delta_0 + \eta f(\eta)c \sum_{t=0}^{\infty} (1-a\eta)^t, \tag{62}$$

$$\le (1-a\eta)^T \delta_0 + \eta f(\eta)c \cdot \frac{1}{c\eta} = (1-a\eta)^T \delta_0 + \frac{c}{a} f(\eta). \tag{63}$$

Therefore, $\delta_T < \epsilon$ if $(1-a\eta)^T < \epsilon/2$ and $f(\eta)c/a < \epsilon/2$. Once can deduce that $(1-a\eta)^T \delta_0 < \epsilon/2$ if $\log(2\delta_0/\epsilon) < T\log(1-a\eta)$, which happens if $\log(2\delta_0/\epsilon) < a\eta T$. $\qquad\square$

## G    Proof of Theorem 18

The functions $f_{i,I}$ are $L$-smooth. This implies that $f_I$ is also $L$-smooth. We can then write

$$f_I(\boldsymbol{w}_{t+1}) \le f_I(\boldsymbol{w}_t) + \nabla f_I(\boldsymbol{w}_t)^\mathsf{T}(\boldsymbol{w}_{t+1} - \boldsymbol{w}_t) + \frac{L}{2}\|\boldsymbol{w}_{t+1} - \boldsymbol{w}_t\|^2, \tag{64}$$

for any $\boldsymbol{w}_t, \boldsymbol{w}_{t+1} \in \mathbb{R}^d$. Substituting them with the stochastic gradient descent updates, *i.e.*, $\boldsymbol{w}_{t+1} = \boldsymbol{w}_t - \eta\nabla f_i(\boldsymbol{w}_t)$, where $i \sim \mathcal{U}([n])$, we obtain

$$f_I(\boldsymbol{w}_{t+1}) \le f_I(\boldsymbol{w}_t) - \eta\nabla f_I(\boldsymbol{w}_t)^\mathsf{T}\nabla f_i(\boldsymbol{w}_t) + \frac{L\eta^2}{2}\|\nabla f_i(\boldsymbol{w}_t)\|^2. \tag{65}$$

Taking conditional expectation with respect to $\boldsymbol{w}_t$, we obtain

$$\mathbb{E}[f_I(\boldsymbol{w}_{t+1}) \mid \boldsymbol{w}_t] \le f_I(\boldsymbol{w}_t) - \eta\,\mathbb{E}_i[\nabla f_I(\boldsymbol{w}_t)^\mathsf{T}\nabla f_i(\boldsymbol{w}_t)] + \frac{L\eta^2}{2}\mathbb{E}_i[\|\nabla f_i(\boldsymbol{w}_t)\|^2]. \tag{66}$$

Using Lemma 27 we get:

$$\mathbb{E}[f_I(\boldsymbol{w}_{t+1}) \mid \boldsymbol{w}_t] \le f_I(\boldsymbol{w}_t) - 2\eta\mu\left(1 - \frac{\eta L^2}{2\mu}\right)(f_I(\boldsymbol{w}_t) - f_I^*) + \eta^2 L^2\Delta_{f_I}$$
$$+ \frac{3}{2}L\eta^2\lambda\frac{1}{n_O}\sum_{i\in n_O}\|\boldsymbol{h}_i(\boldsymbol{o}_i, \boldsymbol{w}_t)\|^2. \tag{67}$$

We assume that $\boldsymbol{w}_t, \boldsymbol{w}_{t+1} \in \mathsf{W}_{\mathrm{SGD}}$. This implies $\frac{1}{n_O}\sum_{i\in n_O}\|\boldsymbol{h}_i(\boldsymbol{o}_i, \boldsymbol{w}_t)\|^2 \le M$. Taking expected value on both sides of (67) and substituting $\delta_t = \mathbb{E}[f_I(\boldsymbol{w}_t) - f_I^*]$ we obtain

$$\delta_{t+1} \le (1 - \eta\mu)\delta_t + \eta^2\left(L^2\Delta_{f_I} + \frac{3}{2}\lambda LM\right) \tag{68}$$

where we have used $\eta < \mu/L^2$ to deduce $2\eta\mu\left(1 - \frac{\eta L^2}{2\mu}\right) > \eta\mu$. Using Lemma 29 establishes the result provided

$$\eta < \min\left\{\frac{1}{\mu}, \frac{\mu}{L^2}, \frac{\epsilon\mu}{3\lambda LM + 2L^2\Delta_{f_I}}\right\}. \tag{69}$$

Noting that $\mu/L \le 1$, this condition reduces to $\eta < \frac{\mu}{L}\min\left\{\frac{1}{L}, \frac{\epsilon}{3\lambda M + 2L\Delta_{f_I}}\right\}$.

## H   Proof of Theorem 19

The functions $f_{i,I}$ are $L$-smooth. This implies that $f_I$ is also $L$-smooth. We can then write

$$f_I(\boldsymbol{w}_{t+1}) \le f_I(\boldsymbol{w}_t) + \nabla f_I(\boldsymbol{w}_t)^{\mathsf{T}}(\boldsymbol{w}_{t+1} - \boldsymbol{w}_t) + \frac{L}{2}\left\|\boldsymbol{w}_{t+1} - \boldsymbol{w}_t\right\|^2, \tag{70}$$

for any $\boldsymbol{w}_t, \boldsymbol{w}_{t+1} \in \mathbb{R}^d$. Substituting them with the adaptive algorithm's updates, *i.e.,* $\boldsymbol{w}_{t+1} = \boldsymbol{w}_t - \eta \sigma'_{c_t}(f_i(\boldsymbol{w}_t))\nabla f_i(\boldsymbol{w}_t)$, where $i \sim \mathcal{U}([n])$, we obtain

$$f_I(\boldsymbol{w}_{t+1}) \le f_I(\boldsymbol{w}_t) - \eta \, \sigma'_{c_t \sigma_{c_t}}(f_i(\boldsymbol{w}_t))\nabla f_I(\boldsymbol{w}_t)^{\mathsf{T}}\nabla f_i(\boldsymbol{w}_t) + \frac{L\eta^2}{2}\sigma'_{c_t}(f_i(\boldsymbol{w}_t))^2 \left\|\nabla f_i(\boldsymbol{w}_t)\right\|^2. \tag{71}$$

Taking conditional expectation with respect to $\boldsymbol{w}_t$, we obtain

$$\mathbb{E}[f_I(\boldsymbol{w}_{t+1}) \,|\, \boldsymbol{w}_t] \le f_I(\boldsymbol{w}_t) - \eta\mathbb{E}_i[\sigma'_{c_t}(f_i(\boldsymbol{w}_t))\nabla f_I(\boldsymbol{w}_t)^{\mathsf{T}}\nabla f_i(\boldsymbol{w}_t)]$$
$$+ \frac{L\eta^2}{2}\mathbb{E}_i[\sigma'_{c_t}(f_i(\boldsymbol{w}_t))^2 \left\|\nabla f_i(\boldsymbol{w}_t)\right\|^2]. \tag{72}$$

Using Lemma 28 we get:

$$\mathbb{E}[f_I(\boldsymbol{w}_{t+1}) \,|\, \boldsymbol{w}_t] \le f_I(\boldsymbol{w}_t) - 2\eta\mu\phi\left(1 - \frac{\eta L^2}{2\mu\phi}\right)(f_I(\boldsymbol{w}_t) - f_I^*) + \eta^2 L^2 \Delta_{f_I}\zeta$$
$$+ \frac{3}{2}L\eta^2\lambda\frac{1}{n_O}\sum_{i \in n_O}\sigma'_{c_t}(f_i(\boldsymbol{w}_t))^2 \left\|\boldsymbol{h}_i(\boldsymbol{o}_i, \boldsymbol{w}_t)\right\|^2. \tag{73}$$

We assume that $\boldsymbol{w}_t, \boldsymbol{w}_{t+1} \in \mathsf{W}_{\text{SGD}}$. This implies $\frac{1}{n_O}\sum_{i \in n_O}\sigma'_{c_t}(f_i(\boldsymbol{w}_t))^2 \left\|\boldsymbol{h}_i(\boldsymbol{o}_i, \boldsymbol{w}_t)\right\|^2 \le M$. Taking expected value on both sides of (76) and substituting $\delta_t = \mathbb{E}[f_I(\boldsymbol{w}_t) - f_I^*]$ we obtain

$$\delta_{t+1} \le (1 - \eta\mu\phi)\delta_t + \eta^2\left(L^2\Delta_{f_I}\zeta + \frac{3}{2}L\eta^2\lambda M\right) \tag{74}$$

where we have used $\eta < \mu\phi/L^2$ to deduce $2\eta\mu\phi\left(1 - \frac{\eta L^2}{2\mu\phi}\right) > \eta\mu\phi$. Using Lemma 29 establishes the result provided

$$\eta < \min\left\{\frac{1}{\mu\phi}, \frac{\mu\phi}{L^2}, \frac{\epsilon\mu\phi}{3\lambda LM + 2L^2\Delta_{f_I}\zeta}\right\}. \tag{75}$$

Noting that $\mu/L \le 1$, this condition reduces to $\eta < \frac{\mu\phi}{L}\min\left\{\frac{1}{L}, \frac{\epsilon}{3\lambda M + 2L\Delta_{f_I}\zeta}\right\}$.

## I   Proof of Theorem 22

Following the same line of arguments as in the proof of Theorem 19 (Appendix H) we get

$$\mathbb{E}[f_I(\boldsymbol{w}_{t+1}) \,|\, \boldsymbol{w}_t] \le f_I(\boldsymbol{w}_t) - 2\eta\mu\phi\left(1 - \frac{\eta L^2}{2\mu\phi}\right)(f_I(\boldsymbol{w}_t) - f_I^*) + \eta^2 L^2 \Delta_{f_I}\zeta$$
$$+ \frac{3}{2}L\eta^2\lambda\frac{1}{n_O}\sum_{i \in n_O}\sigma'_{c_s}(f_i(\boldsymbol{w}_s))^2 \left\|\boldsymbol{h}_i(\boldsymbol{o}_i, \boldsymbol{w}_t)\right\|^2. \tag{76}$$

for $t \in [s, s + T - 1]$ and an $s \in \{0, T, 2T, \ldots\}$. We assume that $\boldsymbol{w}_t, \boldsymbol{w}_{t+1} \in \mathsf{W}_{\sigma-\text{GNC}}$. This implies $\lambda\frac{1}{n_O}\sum_{i \in n_O}\sigma'_{c_s}(f_i(\boldsymbol{w}_s))^2 \left\|\boldsymbol{h}_i(\boldsymbol{o}_i, \boldsymbol{w}_t)\right\|^2$ is bounded by $M$. Taking expected value on both sides of (76) and substituting $\delta_t = \mathbb{E}[f_I(\boldsymbol{w}_t) - f_I^*]$ we obtain

$$\delta_{t+1} \le (1 - \eta\mu\phi)\delta_t + \eta^2\left(L^2\Delta_{f_I}\zeta + \frac{3}{2}L\eta^2\lambda M\right). \tag{77}$$

The result then follows from the same line of argument in Appendix H.

