# OpenReview forum: "Outlier Robust Training of Machine Learning Models"
_TMLR — Rejected by TMLR_

### Review · Reviewer_HNBx · 2025-03-04

**Summary Of Contributions:**

This paper presents a unified framework for robust training of machine learning models in the presence of outliers, making several significant contributions:

- Theoretical Unification: The authors bridge two previously divergent approaches to robust losses (robust estimation and risk-minimization) through a modified Black-Rangarajan duality in Corollary 4, introducing the concept of a "robust loss kernel."

- Algorithmic Innovation: The work develops two complementary algorithm classes—graduated non-convexity and adaptive training algorithms—enhanced with a parameter update rule that intuitively interprets weights as inlier probabilities.

- Empirical Validation: The proposed methods demonstrate exceptional robustness across diverse applications, including regression, classification, and neural scene reconstruction, maintaining performance even with high outlier rates.

**Audience:**

Yes

**Broader Impact Concerns:**

I do not identify any significant ethical concerns or negative societal impacts associated with this paper. The methods for outlier-robust training represent a technical contribution that enhances model performance without introducing notable ethical issues.

**Claims And Evidence:**

Yes

**Requested Changes:**

- The test accuracy in Figure 3(a) is confusing; it would be better to use test RMSE on the y-axis. Additionally, the plots for GD, Adaptive GM, and Adaptive TL become indistinguishable due to poor scaling.

- The experimental setting should be explained in more detail in Section 7.2. For instance, it's unclear in Figure 1 whether the proposed Adaptive Training Algorithm is implemented with SGD or ADAM.

- The paper contains 23 Assumptions/Theorems/Remarks with various algorithm types and variants. While this unifies many algorithms and scenarios, it would be clearer to use algorithm environments and highlight the best algorithm variant with colors.

- There are typos in Appendix A around Equation 34 (such as the unnecessary w). Also, Equation 33 should contain $\Psi(u)$ instead of $\Psi(r)$. Additionally, in Equation 13, the minimization with respect to $u_i$ is unusual.

- In Section 4.1, the proof of Corollary 4 presents two results. The first derives from Black-Rangarajan duality, which appears to require $r > 0$. The authors also provide a detailed first-principles proof that lacks clarity, as mentioned previously. The paper should clarify the conditions when introducing this corollary and its proof.

**Strengths And Weaknesses:**

**Strength**

- The paper effectively bridges the gap between robust estimation methods popular in robotics/computer vision and risk-minimization approaches used in deep learning through a weighted loss formulation that is conceptually clear and intuitive.

- The proposed algorithms demonstrate excellent performance even with extremely high outlier rates (up to 80-90%), significantly outperforming baselines. The empirical validation spans diverse applications (regression, classification, neural scene reconstruction), convincingly demonstrating the general applicability of the approach.

- The theoretical analysis provides rigorous convergence guarantees with comprehensive proofs in the Appendix, establishing a solid mathematical foundation for the methods.

- The paper introduces an intuitive parameter update rule that eliminates the need for manual parameter tuning, enhancing practical usability.

---

**Weakness**

- While the idea of formulating a weighted loss is natural, the technical novelty appears limited. As mentioned on page 7 (after equation 14), the primary innovation involves taking the square root of the cross-entropy loss before applying the formula, which is a relatively straightforward modification.

- The relationship between the proposed algorithms and previous algorithms in the literature remains unclear. The paper would benefit from further clarification and explicit demonstrations of these relationships.

- Although the paper demonstrates robustness to high outlier rates, it fails to establish a clear mathematical relationship between convergence and the fraction of outliers, which limits theoretical understanding of the breakdown point.

- The paper contains several mathematical typos and presentation issues that affect clarity, which I will detail in the Requested Changes section.

---

### Review · Reviewer_czW5 · 2025-04-01

**Summary Of Contributions:**

This paper introduces a unified framework for robust training of machine learning models in the presence of outliers by proposing a modified Black-Rangarajan duality that bridges the gap between robust estimation (common in robotics/vision) and risk minimization (common in deep learning). The key contributions include: (1) a unified robust loss kernel $\sigma$ that generalizes existing robust losses, (2) two novel training algorithms—Graduated Non-Convexity (GNC) and Adaptive Training—equipped with an inlier-probability-based parameter update rule, and (3) theoretical guarantees showing these algorithms converge to outlier-free optima under arbitrary outliers while reducing gradient variance. The framework is validated on regression, classification, and neural scene reconstruction tasks, demonstrating robustness even with 80% outliers, and offers a principled way to design and analyze robust losses across domains.

**Audience:**

Yes

**Claims And Evidence:**

Yes

**Requested Changes:**

1. Could you elaborate on the motivation for why combining robust estimation and risk minimization is necessary especially from application perspective more clearly?
2. Could you add more real-world noisy label tasks like WebVision?
3. y-axis label of Figure 3(a) is not accurate. Maybe change it to "Test RMSE".
4. Two metrics (PSNR and LPIPS) in Figure 4 are unclear. The authors call them "Test accuracy" in the caption, and then I can't see any benefits of the proposed algorithms compared to baselines.
5. Please fix the figure position on page 15.
6. Figure 6 is more straightforward, but the baseline (Adam) is too simple. Could you add Gradient Clipping, Normalized Gradient, and also **adversarial training** as baselines?
7. Do the proposed methods have any computational overhead? Please compare with other baselines.

**Strengths And Weaknesses:**

**Strengths:**
1. The modified Black-Rangarajan duality bridges a gap between robust estimation and risk minimization.
2. Two proposed algorithms,  GNC and Adaptive Training, are supported by convergence guarantees.
3. The authors validate their algorithms in three applications: linear regression, image classification, and neural scene rendering.


**Weaknesses:**
1. The motivation for why combining robust estimation and risk minimization is necessary especially from application perspective is not very clear.
2. The experiments are limited to synthetic or controlled outlier settings, far from real-world settings.
3. Some explanations are not clear or accurate, especially in the Experiments section.

---

### Review · Reviewer_Qqi6 · 2025-04-04

**Summary Of Contributions:**

The paper studies the problem of training machine learning models in the presence of outliers, which is common in real-world applications, including robotics and computer vision. The authors examine two categories of approaches for robust loss design: one from robust estimation (common in robotics and computer vision) and another from the risk minimization framework (common in deep learning).
- They propose a modified Black-Rangarajan duality, which unifies the two views by introducing a robust loss kernel $\sigma$, allowing cross-application of robust loss functions across both domains. This duality transforms robust estimation problems into weighted optimization problems using original loss functions (e.g., cross-entropy), enabling the integration of robust training techniques in training deep neural networks.
- Building on this foundation, the paper introduces two robust training algorithm classes: Graduated Non-Convexity (GNC) and Adaptive Training Algorithms, which adaptively weight training samples based on their inlier probability. The authors provide a novel parameter update rule grounded in conformal prediction theory with no hyperparameter tuning needed. This work derives convergence guarantees for these methods even in the presence of arbitrary (distribution-free) outliers. Extensive theoretical analysis shows that these algorithms expand the region of convergence and reduce gradient variance.
- Empirical results across regression, classification, and neural scene rendering tasks (e.g., Nerfacto) demonstrate the superior performance of the proposed methods. Even with up to 90% outlier data, the proposed algorithms outperform baselines and recover accurate models.

**Audience:**

Yes

**Claims And Evidence:**

No

**Requested Changes:**

- The introduction of the problem of robust estimation is abstract in a first read. It would be better to give a running example in the introduction to help readers understand the problem more.

- Discussion of important literature is postponed to Section 8. It would be better to discuss the most related references in the introduction.

- It would be helpful if the authors could add figures to illustrate the working process of the proposed algorithms.

- Figure in Section 7.3 is cut in between the pages. Please revise it.

**Strengths And Weaknesses:**

### Strengths

- The paper proposes a modified Black-Rangarajan duality that elegantly unifies two major paradigms of robust loss design. This formulation is broadly applicable across loss functions (e.g., MSE, cross-entropy).
- This work designs two algorithms, Graduated Non-Convexity and Adaptive Training Algorithm, for robust loss optimization. They are compatible with SGD, Adam. Also, they require no hand-tuned outlier thresholds.
- The paper conducts comprehensive experiments across three domains—linear regression, image classification, and neural scene reconstruction. For image classification on the CIFAR-10 dataset with symmetric label noise, the robust algorithms consistently outperform standard SGD and other baseline methods, maintaining high accuracy even as the outlier fraction increases up to 90%. In neural radiance field rendering (using the Nerfacto model), the Adaptive TL method successfully recovers meaningful scene reconstructions under extreme pixel-level noise, where vanilla Adam fails.

### Weaknesses
- There are many algorithms have been proposed to improve the model robustness, such as distance-based regularization (Gouk et al.) and Sharpness-aware minimization (Foret et al.). It would be better to compare them in the experiments, especially in training deep neural networks (Section 7.2 and Section 7.3).

- This work lack real-world noisy dataset evaluation. While the paper shows strong performance on synthetic noise (e.g., Gaussian outliers, uniform label corruption), it does not evaluate on real-world noisy datasets like WebVision or Clothing1M, which could better test the framework’s effectiveness in unstructured noise scenarios.

- Although the experiments span three domains, they primarily use standard tasks and shallow models (e.g., linear regression, DLA-34 for CIFAR-10). There’s limited testing on larger-scale architectures, such as VisionTransformers.

---

### Decision · Action_Editor_S3c1 · 2025-06-23

**Recommendation:** Reject

**Audience:**

Yes

**Audience Explanation:**

The general topic of robust training in the presence of outliers is of substantial interest to TMLR’s readership, particularly in areas like vision, robotics, and general deep learning. The attempt to unify robust estimation and risk-minimization frameworks under a common duality lens is intellectually engaging. However, while the core idea may spark interest, the execution and empirical validation fall short of TMLR’s standards.

**Claims And Evidence:**

No

**Claims Explanation:**

While the paper proposes a theoretically interesting unification of robust loss design paradigms through a modified Black-Rangarajan duality, the empirical and comparative support for its effectiveness is insufficient. Many claims, such as the general superiority of the proposed methods, are demonstrated only in synthetic or controlled environments.